# Acetylcholine acts on songbird premotor circuitry to invigorate vocal output

Paul I Jaffe[1,2,3]*, Michael S Brainard[1,2,3,4]*

[1]Departments of Physiology and Psychiatry, University of California, San Francisco, San Francisco, United States; [2]Center for Integrative Neuroscience, University of California, San Francisco, San Francisco, United States; [3]Kavli Institute for Fundamental Neuroscience, University of California, San Francisco, San Francisco, United States; [4]Howard Hughes Medical Institute, University of California, San Francisco, San Francisco, United States

**Abstract** Acetylcholine is well-understood to enhance cortical sensory responses and perceptual sensitivity in aroused or attentive states. Yet little is known about cholinergic influences on motor cortical regions. Here we use the quantifiable nature of birdsong to investigate how acetylcholine modulates the cortical (pallial) premotor nucleus HVC and shapes vocal output. We found that dialyzing the cholinergic agonist carbachol into HVC increased the pitch, amplitude, tempo and stereotypy of song, similar to the natural invigoration of song that occurs when males direct their songs to females. These carbachol-induced effects were associated with increased neural activity in HVC and occurred independently of basal ganglia circuitry. Moreover, we discovered that the normal invigoration of female-directed song was also accompanied by increased HVC activity and was attenuated by blocking muscarinic acetylcholine receptors. These results indicate that, analogous to its influence on sensory systems, acetylcholine can act directly on cortical premotor circuitry to adaptively shape behavior.

*For correspondence:
pauljaffe7@gmail.com (PIJ);
msb@phy.ucsf.edu (MSB)

## Introduction

Physiological arousal is accompanied by global changes in brain state that facilitate sensory processing and enable rapid behavioral responses (*Lee and Dan, 2012*). In the sensory domain, active and aroused behavioral states are associated with enhanced perceptual capabilities that can aid detection and processing of threats or other salient stimuli (*Bennett et al., 2013*; *McGinley et al., 2015*; *Woods et al., 2013*). Analogously, in the motor domain, greater arousal can enable more rapid, forceful, and precise movements—that is, greater vigor—which work in tandem with enhanced sensory processing to coordinate adaptive behavioral responses (*Bouman et al., 2015*; *DiGirolamo et al., 2016*; *Lovett-Barron et al., 2017*; *McGinley et al., 2015*). Acetylcholine, which figures prominently in the ascending arousal system, has been linked to the arousal-related enhancement of sensory processing by direct action on sensory cortices (*Fu et al., 2014*; *Herrero et al., 2008*; *St Peters et al., 2011*; *Pinto et al., 2013*; *Reimer et al., 2016*). However, while motor cortical regions receive dense cholinergic innervation from the nucleus basalis (NBM; *Eckenstein et al., 1988*; *McKinney et al., 1983*; *Raghanti et al., 2008*), the extent to which cholinergic signaling in cortex contributes to motor invigoration observed in aroused behavioral states remains unknown.

Previous work on the control of movement vigor has focused primarily on dopaminergic signaling in the basal ganglia, which appears to be particularly important for invigorating movements in order to obtain reward and in other motivational contexts (*Berke, 2018*; *Dudman and Krakauer, 2016*; *Turner and Desmurget, 2010*). Nevertheless, experimental and pathological disturbances of the cholinergic system point to a possible cholinergic contribution to the invigoration of motor output. Indeed, lesions of NBM and diseases of the cholinergic system can be accompanied by a reduction

in the speed, force, and amplitude of movements (*Berger-Sweeney et al., 1994*; *Buchman et al., 2007*; *Ferris and Farlow, 2013*; *Goldman et al., 1999*), in addition to more general impairments in motor skill acquisition and motor recovery following cortical lesions (*Conner et al., 2003*; *Conner et al., 2005*; *Conner et al., 2010*). Moreover, stimulation of NBM can invigorate movements of rat vibrissae that are induced by stimulation of motor cortex (*Berg et al., 2005*). These observations raise the question of whether acetylcholine can adaptively modulate movements by direct action on cortical circuitry, and whether its influence is separable from the control of vigor by dopaminergic signaling in the basal ganglia.

Birdsong is an attractive system for evaluating cholinergic contributions to the control of motor vigor in states of elevated arousal. Song is a learned and readily quantifiable motor skill that is controlled by well-defined cortical and basal ganglia circuitry. Like speech, song is naturally produced in states of greater or lesser arousal that are associated with changes in 'vocal vigor'. In particular, female-directed song during courtship is associated with greater pitch and tempo, altered song amplitude, and increased acoustic stereotypy compared to the undirected song that male birds sing in isolation (*Cooper and Goller, 2006*; *James and Sakata, 2015*; *Kao et al., 2005*; *Sakata et al., 2008*; *Sossinka and Böhner, 1980*; *Suri and Rajan, 2018*). Consistent with the notion that directed song reflects a state of greater physiological arousal, pre-song heart rate is faster for directed song than undirected song (*Cooper and Goller, 2006*). Moreover, the cortical premotor song nucleus HVC, like mammalian motor cortex, receives strong cholinergic innervation from the basal forebrain (*Ryan and Arnold, 1981*; *Zuschratter and Scheich, 1990*), and cholinergic manipulations can alter excitability of HVC neurons in vitro and modulate their responses to song playback in vivo in anesthetized birds (*Shea and Margoliash, 2003*; *Shea et al., 2010*). However, as in mammalian systems, the extent to which cholinergic action on cortical motor regions contributes to arousal-related changes in behavior has not been examined.

In this study, we use the quantifiable nature of birdsong to investigate how cholinergic signaling modulates the cortical (pallial) premotor nucleus HVC, which has similar cell types, connections, and function to mammalian motor cortical regions. Using a combination of electrophysiological recordings and targeted pharmacological manipulations in singing birds, we show that activation of cortical acetylcholine receptors leads to elevated HVC activity accompanied by an increase in the pitch, amplitude, tempo, and stereotypy of undirected song. Moreover, we demonstrate that the normal invigoration of song that occurs in the presence of a female bird is also associated with elevated HVC activity, and that this invigoration can be attenuated by blocking muscarinic acetylcholine receptors in HVC. Strikingly, song invigoration in response to increased cholinergic tone in HVC persists even when contributions of basal ganglia circuitry to song are pharmacologically blocked. Overall, our findings argue that acetylcholine can act directly on cortical premotor circuitry to adaptively shape behavioral outputs in aroused behavioral states, and indicate that the control of movement vigor in central circuits is more distributed than previously appreciated.

## Results

### Acetylcholine invigorates song and increases song stereotypy

To determine whether cholinergic modulation of cortical motor areas can shape behavioral output, we evaluated changes to the acoustic structure of undirected song ('baseline song') following localized delivery of the cholinergic agonist carbachol into HVC of adult male Bengalese finches (*Lonchura striata domestica*; *Figure 1A*, n = 8 birds; see Methods). Carbachol did not cause gross distortions of the individual acoustic elements of the song, referred to as 'syllables' (*Figure 1B*). However, quantitative analysis revealed that carbachol had a number of consistent effects on syllable structure that largely paralleled those observed during female-directed song, including increases in pitch, syllable stereotypy, tempo, and amplitude (*Cooper and Goller, 2006*; *James and Sakata, 2015*; *Kao et al., 2005*; *Sakata et al., 2008*; *Sossinka and Böhner, 1980*; *Suri and Rajan, 2018*).

To quantify effects on pitch, we identified syllables with well-defined harmonic structure and computed the normalized fundamental frequency of each syllable (drug/baseline) in a two-hour period during carbachol infusion relative to a baseline period prior to drug infusion (see Methods). To control for possible circadian fluctuations in behavior (*Chi and Margoliash, 2001*; *Derégnaucourt et al., 2005*; *Glaze and Troyer, 2006*; *Wood et al., 2013*), we compared the

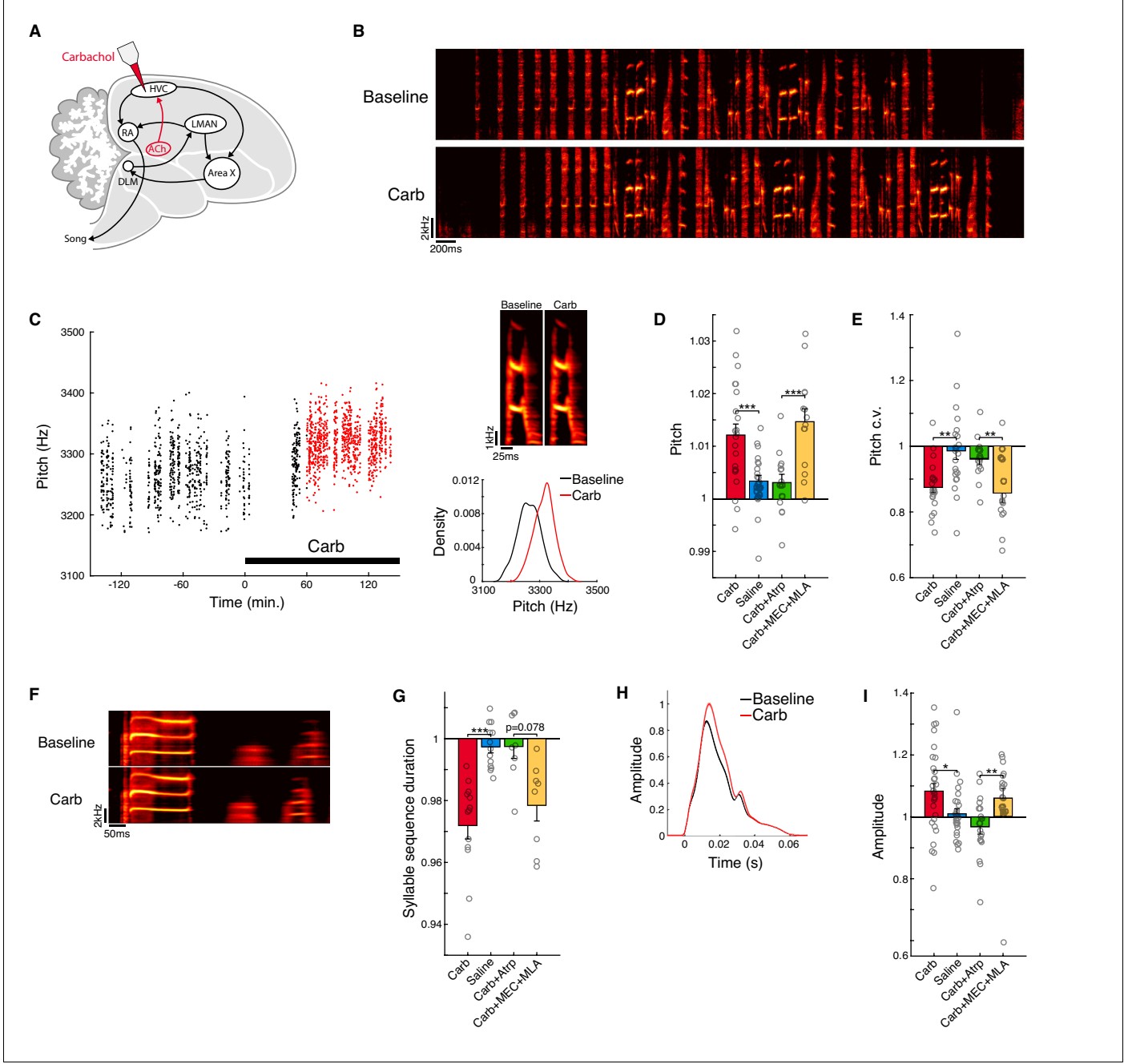

**Figure 1.** Activation of muscarinic receptors in HVC increases motor vigor. (**A**) Experiment schematic and song system. Carbachol (Carb) was microdialyzed into HVC. HVC receives a cholinergic projection from a homolog of the nucleus basalis (ACh, red). The avian song control system consists of a direct motor pathway that includes the cortical nuclei HVC (used as a proper name) and RA (the robust nucleus of the arcopallium), which projects to the brainstem premotor regions that control vocal musculature; and an Anterior Forebrain Pathway (AFP) that includes the basal ganglia homologue Area X, the thalamic nucleus DLM (the dorsolateral nucleus of the medial thalamus), and the cortical nucleus LMAN (the lateral magnocellular nucleus of the anterior nidopallium), which projects back to the motor pathway at the level of RA. (**B**) Example song bouts before and after carbachol. (**C**) Change in pitch produced by carbachol for one example syllable (one experiment). Left, time course of raw pitch values. Red points indicate data used for analysis of carbachol effects (60–180 min. following onset of dialysis). Top right, rendition-averaged spectrograms. Lower right, pitch distributions during baseline (black) and Carb (red) conditions. (**D**) Normalized (drug/baseline) pitch (mean ± s.e.m. increase in pitch for Carb: 1.2 ± 0.21%, n = 20 experiments; Saline: 0.34 ± 0.11%, n = 17 experiments; Carb+Atrp: 0.31 ± 0.16%, n = 10 experiments; Carb+MEC+MLA: 1.5 ± 0.24%, n = 10 experiments; Carb vs. Saline, p=0.00088, two-tailed signed-rank test, n = 22 syllables, eight birds; Carb+Atrp vs. Carb+MEC+MLA, p=0.00024, two-tailed signed-rank test, n = 14 syllables, five birds). (**E**) Normalized (drug/baseline) pitch c.v. (mean ± s.e.m. reduction in pitch c.v. for Carb: 13 ± 1.6%, n = 20 experiments; Saline: 1.4 ± 2.7%, n = 17 experiments; Carb+Atrp: 4.0 ± 1.7%, n = 10 experiments; Carb+MEC+MLA: 14 ± 2.9%, n = 10

*Figure 1 continued on next page*

**Figure 1 continued**

experiments; Carb vs. Saline, p=0.0014, two-tailed signed-rank test, n = 22 syllables, eight birds; Carb+Atrp vs. Carb+MEC+MLA, p=0.0023, two-tailed signed-rank test, n = 14 syllables, five birds). (F) Rendition-averaged spectrograms of one example syllable sequence before and after carbachol (one experiment). (G) Normalized (drug/baseline) syllable sequence duration (mean ± s.e.m. reduction in sequence duration for Carb: 2.8 ± 0.43%, n = 20 experiments; Saline: 0.26 ± 0.20%, n = 15 experiments; Carb+Atrp: 0.25 ± 0.39%, n = 9 experiments; Carb+MEC+MLA: 2.2 ± 0.51%, n = 10 experiments; Carb vs. Saline, p=0.00024, two-tailed signed-rank test, n = 13 syllable sequences, eight birds; Carb+Atrp vs. Carb+MEC+MLA, p=0.078, two-tailed signed-rank test, n = 8 syllable sequences, five birds). (H) Amplitude envelopes (mean ± s.e.m.) for one example syllable before and after carbachol (one experiment). Amplitude envelopes were normalized to the maximum value in the carbachol condition. (I) Normalized (drug/baseline) amplitude (mean ± s.e.m. increase in amplitude for Carb: 8.3 ± 2.5%, n = 18 experiments; Saline: 0.99 ± 1.6%, n = 17 experiments; Carb+Atrp: −3.2 ± 2.4%, n = 10 experiments; Carb+MEC+MLA: 6.1 ± 3.1%, n = 9 experiments; Carb vs. Saline, p=0.012, two-tailed signed-rank test, n = 30 syllables, eight birds; Carb +Atrp vs. Carb+MEC+MLA, p=0.0033, two-tailed signed-rank test, n = 18 syllables, five birds). ***p<0.001, **p<0.01, *p<0.05. For panels D, E, G, and I, each point represents one syllable or syllable sequence averaged over experiments; Atrp = the muscarinic antagonist atropine; MEC and MLA = the nicotinic antagonists mecamylamine and methyllycaconitine.

The online version of this article includes the following source data and figure supplement(s) for figure 1:

**Source data 1.** Linear mixed effects model analysis of the behavioral effects of carbachol.
**Source data 2.** Source data for the summary analyses in figure panels D, E, G, and I.
**Figure supplement 1.** Further characterization of the tempo and amplitude effects of carbachol.

magnitude of changes following carbachol infusion to the magnitude of changes in response to control saline infusion on alternate days. Compared to saline, carbachol elicited significant increases in pitch (*Figure 1C and D*; increase in pitch for carbachol: 1.2 ± 0.21%, mean ± s.e.m.; saline: 0.34 ± 0.11%; carbachol vs. saline, p=8.8e-4, signed-rank test; see *Figure 1—source data 1* for mixed effects model analysis of behavioral effects).

To quantify effects on syllable stereotypy, we measured changes to the across-rendition coefficient of variation (c.v.) of pitch as in previous studies (*Hampton et al., 2009*; *Kao et al., 2005*). Increases in stereotypy of movements have often been observed in conjunction with increases in motor vigor in other systems (*Manohar et al., 2015*; *Summerside et al., 2018*), and across-rendition stereotypy of acoustic structure is higher during directed song relative to undirected song (*Kao et al., 2005*; *Leblois et al., 2010*; *Sakata et al., 2008*; *Stepanek and Doupe, 2010*). Carbachol produced significant increases in stereotypy as measured by reduced pitch c.v. (*Figure 1C and E*; reduction in pitch c.v. for carbachol: 13 ± 1.6%, mean ± s.e.m.; saline: 1.4 ± 2.7%; carbachol vs. saline, p=0.0014, signed-rank test; *Figure 1—source data 1*).

To quantify effects on song tempo, we calculated changes to the mean duration of defined syllable sequences in each bird's repertoire (e.g. the sequence of three syllables illustrated in *Figure 1F*, see Methods). Carbachol elicited robust increases in tempo as measured by reduced durations of syllable sequences (*Figure 1F and G*; decrease in sequence duration for carbachol: 2.8 ± 0.43%, mean ± s.e.m.; saline: 0.26 ± 0.20%; carbachol vs. saline, p=2.4e-4, signed-rank test; *Figure 1— source data 1*). These increases in tempo reflected a reduction in durations of both syllables and the gaps between syllables; moreover, consistent with the effects of temperature manipulations of HVC (*Long and Fee, 2008*; *Zhang et al., 2017*), we observed a proportionately greater effect for gaps than for syllables (*Figure 1—figure supplement 1*; decrease in syllable duration for carbachol: 0.93 ± 0.58%, decrease in gap duration: 3.2 ± 1.0%; syllables vs. gaps: p=0.085, rank-sum test).

To quantify effects on syllable amplitude, we averaged the smoothed amplitude envelope over the middle 80% of each syllable (see Methods). We found that carbachol significantly increased song amplitude (*Figure 1H and I*, and *Figure 1—figure supplement 1*; increase in amplitude for carbachol: 8.3 ± 2.5%, mean ± s.e.m.; saline: 0.99 ± 1.6%; carbachol vs. saline, p=0.012, signed-rank test; *Figure 1—source data 1*). An increase in amplitude was observed not only for 'harmonic stack' syllables that could be analyzed for changes to pitch, but also for syllables with more varied spectrotemporal properties, including 'sweeps' and 'complex' syllables that had frequency-modulated harmonic components, and 'noisy' syllables that had high spectral entropy (*Figure 1—figure supplement 1*).

In addition to examining effects of carbachol on the acoustic structure of syllables, we also tested whether carbachol dialyzed into HVC alters the sequencing of syllables. This was motivated by prior observations that syllable sequencing is systematically altered in directed song relative to undirected song (*Hampton et al., 2009*; *Sakata et al., 2008*; *Sossinka and Böhner, 1980*). Consistent with the changes to sequencing observed during directed song, carbachol significantly altered transition

probabilities at variable transitions in song ('branch points') (*Figure 2A–C*; n = 15 branch points, seven birds), and significantly increased the number of times that syllables were repeated (*Figure 2D and E*; carbachol vs. baseline, p=0.031, signed-rank test, n = 7 repeated syllables, six birds; *Hampton et al., 2009*; *Sakata et al., 2008*). Further, the effect on syllable repetitions was greater for more variable repetitions, as previously observed for temperature manipulations of HVC (*Figure 2F*; *Zhang et al., 2017*). Previous studies in Bengalese finches have also reported that syllable sequencing can be more stereotyped during directed song, as measured by a reduction in transition entropy at variable transitions in song (*Hampton et al., 2009*; *Sakata et al., 2008*). Consistent with this, we observed modest trends toward reduced transition entropy for both branch points and repeated syllables (see Methods; mean ± s.e.m. reduction in transition entropy for branch points: 7.1 ± 6.3%; repeated syllables: 5.3 ± 2.5%; carbachol vs. baseline for branch points: p=0.17, repeated syllables: p=0.078, signed-rank test). Hence, carbachol dialyzed into HVC and directed song were associated with similar changes to syllable sequencing.

To determine whether the effects of carbachol are mediated by muscarinic or nicotinic receptors, both of which are expressed in HVC (*Asogwa et al., 2018*; *Ball et al., 1990*; *Watson et al., 1988*), we tested how the effects produced by carbachol were affected by the concurrent dialysis of either muscarinic or nicotinic receptor antagonists into HVC (n = 5 birds). The effects produced by carbachol on pitch, pitch c.v., amplitude, and tempo were attenuated by the muscarinic antagonist atropine, but not by the nicotinic antagonists MEC (mecamylamine) and MLA (methyllycaconitine; *Figures 1D, E, G and I*; carbachol+atropine vs. carbachol+MEC+MLA, p<0.05 for all features except for tempo, signed-rank test; *Figure 1—source data 1*). In summary, we found that activation of muscarinic acetylcholine receptors in HVC enhances song vigor, producing a suite of behavioral changes that largely parallels those observed during directed song (*Cooper and Goller, 2006*; *James and Sakata, 2015*; *Kao et al., 2005*; *Sakata et al., 2008*; *Sossinka and Böhner, 1980*; *Suri and Rajan, 2018*).

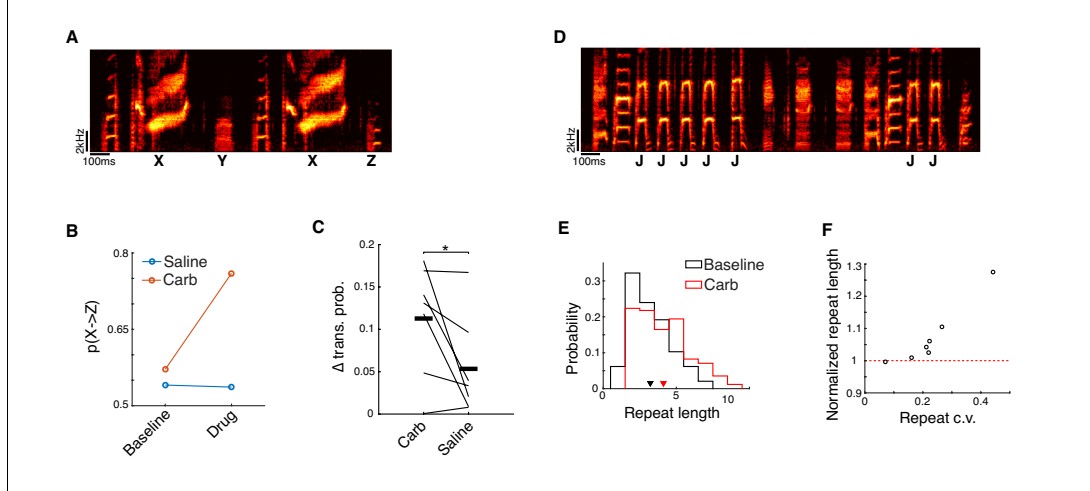

**Figure 2.** Microdialysis of carbachol into HVC alters song sequencing. (**A**) Spectrogram of a song with a divergent branch point. Syllable 'X' can transition to syllable 'Y' or syllable 'Z'. (**B**) Transition probabilities before and after either carbachol or saline dialysis for the branch point shown in panel A. (**C**) Change in transition probability averaged across all branch points for each bird (see Methods; mean ± s.e.m. change for Carb: 0.11 ± 0.025, Saline: 0.053 ± 0.022; Carb vs. Saline, p=0.047, two-sided signed-rank test, n = 15 branch points, seven birds, 11 experiments for Carb, 12 experiments for Saline). On a case-by-case basis, transition probabilities at 6 out of 15 branch points were significantly affected by carbachol, while only 1 out of 15 was significantly affected by saline (p<0.05, generalized likelihood ratio test for homogeneity, see Methods). (**D**) Song spectrogram depicting a variably repeated syllable (syllable 'J'). (**E**) Histogram of repeat counts before and after carbachol for the repeated syllable shown in panel D. (**F**) Scatter plot of repeat length c.v. vs. normalized repeat count (carbachol/baseline) after carbachol (p=0.0012, test for non-zero Pearson's correlation coefficient, corr. coeff. = 0.95; mean ± s.e.m. increase in repeat length: 7.4 ± 3.6%; mean ± s.e.m. repeat length c.v.: 0.23 ± 0.043, n = 7 repeated syllables, six birds, 12 experiments). *p<0.05.

The online version of this article includes the following source data for figure 2:

**Source data 1.** Source data for the summary analyses in figure panels C and F.

## Acetylcholine invigorates song via the cortical-brainstem motor pathway

In principle, acetylcholine could invigorate song through either of two major pathways emanating from HVC: through the direct cortical-brainstem motor pathway, via a projection from HVC to RA, or through basal ganglia circuitry, via an indirect pathway from HVC→Area X→DLM→LMAN→RA (the Anterior Forebrain Pathway or 'AFP'; *Figure 1A*). Previous studies in mammalian systems have identified the basal ganglia as a key locus for the modulation of motor vigor (*Panigrahi et al., 2015*; *Yttri and Dudman, 2016*), supporting the possibility that the effects of carbachol infusion into HVC could reflect primarily an influence on AFP circuitry. Further, a number of studies in songbirds have identified the AFP as a critical site for social modulation of pitch variability that occurs during directed song (*Hampton et al., 2009*; *Jarvis et al., 1998*; *Kao and Brainard, 2006*; *Kao et al., 2005*; *Leblois et al., 2010*; *Stepanek and Doupe, 2010*; *Teramitsu and White, 2006*; *Woolley, 2016*).

To determine whether basal ganglia circuitry contributes to acoustic changes produced by carbachol, we microdialyzed carbachol into HVC while inactivating LMAN with muscimol (*Figure 3A and B*; n = 4 birds). LMAN is the main output nucleus of the AFP, and muscimol inactivation of LMAN disconnects the AFP from the song motor pathway (*Andalman and Fee, 2009*; *Ölveczky et al., 2011*; *Stepanek and Doupe, 2010*; *Warren et al., 2011*). As in previous studies, infusion of muscimol caused a significant decrease in pitch variability, confirming that LMAN was effectively inactivated (reduction in pitch c.v.: 23 ± 5.3%, mean ± s.e.m.; muscimol vs. baseline, p=0.0020, signed-rank test). However, even when LMAN was inactivated, infusion of carbachol into HVC caused increases in pitch, pitch stereotypy, tempo, and amplitude (*Figure 3B–F*). For each of these features, the increases produced by combined carbachol + LMAN inactivation were greater than for LMAN inactivation alone (p<0.05 for pitch, amplitude, and tempo; p=0.084 for pitch c.v.; signed-rank test; see also *Figure 3—source data 1* for mixed effects analysis). Moreover, for each feature, the changes elicited by combined carbachol + LMAN inactivation were not significantly different from the sum of the individual effects of carbachol and LMAN inactivation (pitch: p=0.77, pitch c.v.: p=0.85, tempo: p=0.22, amplitude: p=0.71, signed-rank test; *Figure 3—source data 1*). These results indicate that increased cholinergic tone in HVC can modulate song via primary motor circuitry independently of input from the songbird basal ganglia, and thus provide support for a recent model proposing that basal ganglia and non-basal ganglia pathways can make independent and additive contributions to behavior (*Yttri and Dudman, 2018*).

## Acetylcholine increases neural activity in HVC

To determine how increased cholinergic tone alters HVC activity, we recorded multi-unit neural activity in HVC of singing Bengalese finches before and after microdialysis of carbachol (*Figure 4A*). We focused on multi-unit activity, since stable recordings of isolated single units are difficult to maintain over the course of pharmacological manipulations as required for these experiments. The signal-to-noise ratio (SNR) of these recordings ranged from 3.2 to 6.7 (see Methods), and firing rates ranged from 50 to 560 Hz, indicating that most recordings sampled from multiple neurons simultaneously (*Figure 4—figure supplement 1*). Moreover, due to the higher firing rates of inhibitory interneurons relative to the excitatory projection neurons in HVC, such multi-unit activity is likely to primarily reflect interneuron activity (*Hahnloser et al., 2002*; *Kozhevnikov and Fee, 2007*; *Liberti et al., 2016*; *Rauske et al., 2003*).

For each recording site and syllable, we computed rendition-averaged firing rates during saline and carbachol blocks aligned to syllable onsets (*Figure 4B*). During control saline blocks, there was no change in average firing rates relative to baseline, indicating that multiunit recordings remained stable over the period of drug dialysis (*Figure 4C*; change in firing rate in 100 ms window centered on syllable onsets: 1.1 ± 1.8%, mean ± s.e.m.; saline vs. baseline, p=0.64, signed-rank test; n = 118 multi-unit sites x syllables, five birds). In contrast, we found that carbachol significantly increased average firing rates in HVC relative to baseline (*Figure 4D*; increase in firing rate in 100 ms window centered on syllable onsets: 9.9 ± 1.4%, mean ± s.e.m.; carbachol vs. baseline, p=2.8e-9, signed-rank test; n = 202 multi-unit sites x syllables, five birds; see also *Figure 4—source data 1* for mixed effects analysis). In general, the firing rate changes caused by carbachol were complex, varying in magnitude at different time points in song (*Figure 4B* and *Figure 4—figure supplement 1*).

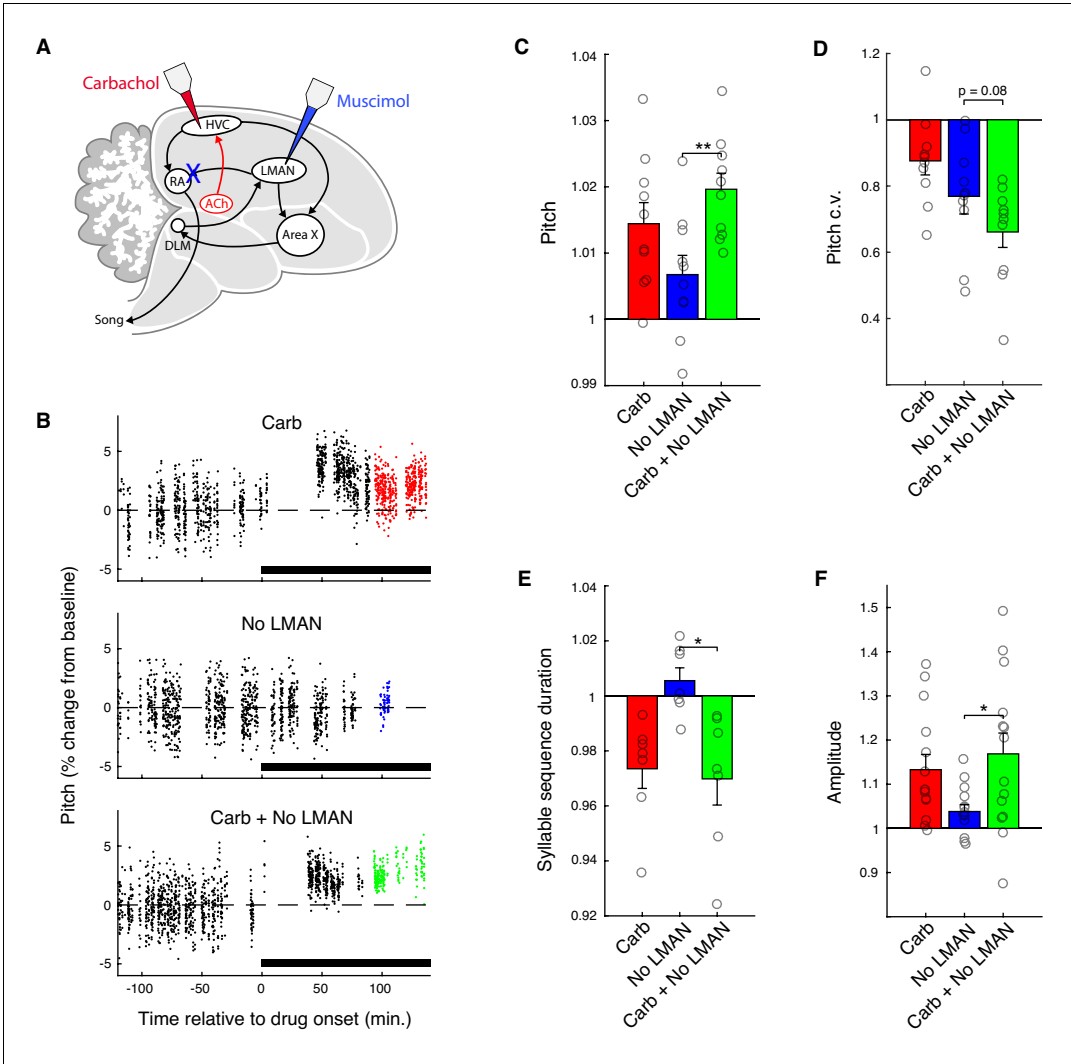

**Figure 3.** Carbachol invigorates song via the cortical-brainstem motor pathway. (**A**) Experiment schematic. Carbachol (Carb) was microdialyzed into HVC, and LMAN was inactivated with muscimol (No LMAN), concurrently or separately. (**B**) Time course of pitch values for representative experiments from one bird (the same syllable is shown for each condition). For each condition, pitch is plotted as percent change relative to the average value during baseline (−120 to 0 min.). Points in color indicate data used for analysis of drug effects (90 to 150 min.). (**C**) Normalized (drug/baseline) pitch (mean ± s.e.m. increase in pitch for Carb: 1.4 ± 0.32%, No LMAN: 0.68 ± 0.29%, Carb+No LMAN: 2.0 ± 0.24%; Carb+No LMAN vs. No LMAN, p=0.0020, two-tailed signed-rank test, n = 10 syllables, four birds). (**D**) Normalized (drug/baseline) pitch c.v. (mean ± s.e.m. reduction in pitch c.v. for Carb: 12 ± 4.2%, No LMAN: 23 ± 5.3%, Carb+No LMAN: 34 ± 4.7%; Carb+No LMAN vs. No LMAN, p=0.084, two-tailed signed-rank test, n = 10 syllables, four birds). (**E**) Normalized (drug/baseline) syllable sequence duration (mean ± s. e.m. reduction in sequence duration for Carb: 2.7 ± 0.72%, No LMAN: −0.55 ± 0.47%, Carb+No LMAN: 3.0 ± 0.95%; Carb+No LMAN vs. No LMAN, p=0.031, two-tailed signed-rank test, n = 7 syllable sequences, four birds). (**F**) Normalized (drug/baseline) amplitude (mean ± s.e.m. increase in amplitude for Carb: 13 ± 3.5%, No LMAN: 3.8 ± 1.5%, Carb+No LMAN: 17 ± 4.7%; Carb+No LMAN vs. No LMAN, p=0.035, two-tailed signed-rank test, n = 14 syllables, four birds). **p<0.01, *p<0.05. For panels C–F, each point represents one syllable or syllable sequence averaged over experiments. For panels C, D, and E, n = 9 experiments for all conditions. For panel F, n = 8 experiments for Carb, nine for No LMAN, and nine for Carb + No LMAN.

The online version of this article includes the following source data for figure 3:

**Source data 1.** Linear mixed effects model analysis of combined carbachol and LMAN inactivation experiments.
**Source data 2.** Source data for the summary analyses in figure panels C−F.

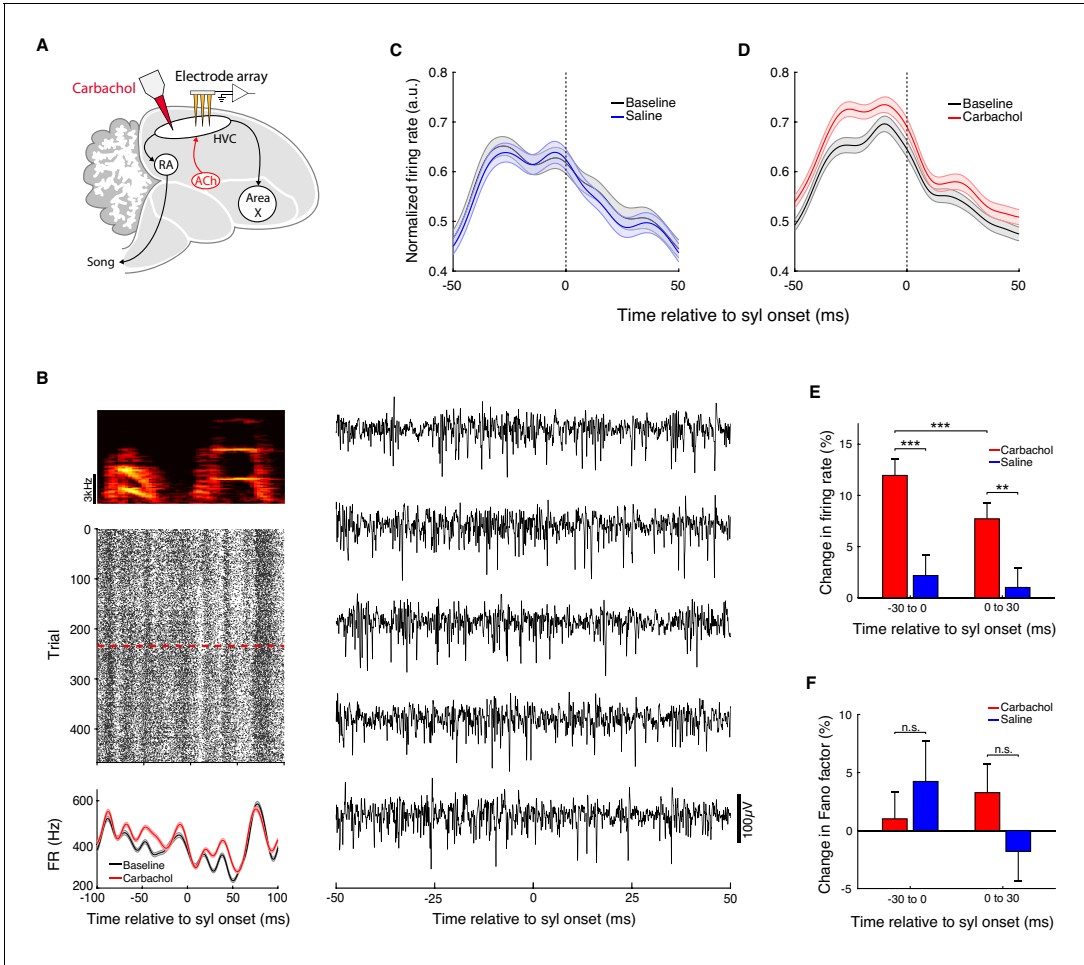

**Figure 4.** Carbachol increases HVC multi-unit firing rates. (A) Experiment schematic. Carbachol was microdialyzed into HVC while recording neural activity with multi-electrode arrays. (B) Example multi-unit site (one experiment; activity aligned to the onset of one syllable). Top left, spectrogram of the syllable used for alignment. Middle left, raster plot of the multi-unit site (red dashed line: onset of carbachol). Bottom left, rendition-averaged firing rates (mean ± s.e.m., smoothed with a 5 ms SD gaussian kernel). Right, example raw traces for this multi-unit site (bandpass filtered between 300 and 4000 Hz). (C) Population average firing rates aligned to syllable onsets, before and after saline. Prior to averaging across sites/syllables, mean firing rates from both baseline and saline blocks were normalized by the maximum of both conditions in a 100 ms window centered on the syllable onset (n = 118 multi-unit sites x syllables, five birds, eight experiments). (D) Population average firing rates aligned to syllable onsets, before and after carbachol. Data are normalized as in panel C (n = 202 multi-unit sites x syllables, five birds, eight experiments). (E) Percent change in firing rate after switch to carbachol relative to baseline, or after switch to saline relative to baseline (mean ± s.e.m. increase in firing rate for Carb in −30 to 0 ms window: 12 ± 1.6%; Carb 0 to 30 ms: 7.7 ± 1.6%, Saline −30 to 0 ms: 2.2 ± 2.0%, Saline 0 to 30 ms: 1.0 ± 1.9%; Carb vs. Saline in −30 to 0 ms window, p=1.3e-4, two-tailed rank-sum test; Carb vs. Saline 0 to 30 ms, p=0.0039, two-tailed rank-sum test; Carb −30 to 0 ms vs. Carb 0 to 30 ms, p=5.4e-4, two-tailed signed-rank test). (F) Percent change in Fano factor after switch to carbachol relative to baseline, or after switch to saline relative to baseline (mean ± s.e.m. change in Fano factor for Carb in −30 to 0 ms window: 1.0 ± 2.3%; Carb 0 to 30 ms: 3.3 ± 2.5%, Saline −30 to 0 ms: 4.2 ± 3.5%, Saline 0 to 30 ms: −1.8 ± 2.5%; Carb vs. Saline in −30 to 0 ms window, p=0.73, two-tailed rank-sum test; Carb vs. Saline 0 to 30 ms, p=0.29, two-tailed rank-sum test). ***p<0.001, **p<0.01, n.s., not significant.

The online version of this article includes the following source data and figure supplement(s) for figure 4:

**Source data 1.** Linear mixed effects model analysis of multi-unit firing rate changes following microdialysis of carbachol or saline.
**Source data 2.** Source data for the summary analyses in figure panels C−F.
**Figure supplement 1.** Additional characterization of multi-unit recordings from microdialysis experiments.

However, on average there was a significant increase in firing rate both preceding and following syllable onsets, with a modestly greater effect in the premotor window preceding syllable onsets (*Figure 4D and E*; maximum of 16% increase in firing rate at 25 ms before syllable onsets, minimum of 6.1% at 15 ms after syllable onsets).

The increased acoustic stereotypy we observed during microdialysis of carbachol led us to consider whether this was caused by a corresponding reduction in neural variability in HVC. To evaluate this, we measured the Fano factor (across-rendition spike count variance/mean spike count) for each multi-unit site and syllable (*Figure 4F*). On average, carbachol did not produce a significant change in Fano factor (carbachol vs. saline, p>0.05 in 30 ms window preceding syllable onsets and 30 ms window after syllable onsets, rank-sum test). We also evaluated neural variability by calculating the spike count variance. We found no significant effect of carbachol relative to saline in the 30 ms window prior to syllable onsets (p=0.17, rank-sum test), and a tendency for carbachol to increase neural variability in the 30 ms window after syllable onsets (p=0.023, rank-sum test). Thus, the increased pitch stereotypy caused by carbachol cannot be explained by reduced neural variability in HVC at the multi-unit level, though we cannot rule out the possibility that this increased behavioral stereotypy arises in part from reduced variability specifically among HVC projection neurons.

## HVC activity is modulated by social context

Directed song and microdialysis of carbachol into HVC were associated with similar behavioral changes, yet previous studies have found only limited evidence that activity within HVC differs between undirected and directed song (*Jarvis et al., 1998*; *Matheson et al., 2016*; *Woolley et al., 2014*). We therefore wondered if we could detect any changes to neural activity in HVC during directed song similar to the increases in activity caused by carbachol infusion. To address this, we recorded multi-unit neural activity in HVC during interleaved blocks of directed and undirected singing (*Figure 5A*; n = 5 birds). We computed rendition-averaged firing rates aligned to syllable onsets separately for directed and undirected songs. The pattern of neural modulation with respect to song features was largely conserved across social contexts: the mean ± s.e.m. correlation coefficient between directed and undirected firing rates was 0.90 ± 0.010 (e.g. *Figure 5B* and *Figure 5—figure supplement 1*; n = 151 multi-unit sites x syllables, 15 unique multi-unit sites, five birds). Nevertheless, HVC multi-unit firing rates were consistently higher during directed song relative to undirected song (*Figure 5B*—5D and *Figure 5—figure supplement 1*; increase in firing rate in 100 ms window centered on syllable onsets: 7.2 ± 1.0%, mean ± s.e.m.; directed vs. undirected, p=1.1e-15, signed-rank test; n = 151 multi-unit sites x syllables, five birds; see also *Figure 5—source data 1* for mixed effects analysis). Further, the percent increase in firing rate for directed song was not significantly different from that observed after microdialysis of carbachol (p=0.90, rank-sum test). The patterns of increased firing during directed song were idiosyncratic across recording sites and syllables (*Figure 5—figure supplement 1*). However, as observed for carbachol, there was on average a significant increase in firing rate both preceding and following syllable onsets (*Figure 5D*; mean ± s.e.m. increase in −30 to 0 ms window: 7.9 ± 1.2%; 0 to 30 ms window: 7.5 ± 1.2%). Overall, these results reveal that both microdialysis of carbachol and directed song are associated with greater activity in HVC, and raise the question of whether cholinergic signaling in HVC normally contributes to social modulation of song.

## Acetylcholine contributes to the social modulation of song

To test whether cholinergic signaling in HVC contributes to the social modulation of song, we quantified how the normal differences between directed and undirected song were affected by dialysis of the muscarinic antagonist atropine into HVC (*Figure 6A and B*). Under control conditions, as previously reported (*Hampton et al., 2009*; *James and Sakata, 2015*; *Sakata et al., 2008*), directed song was higher in pitch, less variable in pitch, and faster than undirected song (*Figure 6D–F*, saline conditions). The increase in song tempo during directed song reflected decreases in the durations of both syllables and gaps, with a proportionately larger decrease in gaps as also observed for carbachol (*Figure 6—figure supplement 1*; mean ± s.e.m. reduction in duration for syllables: 1.0 ± 0.49%, gaps: 4.2 ± 1.4%; syllables vs. gaps, p=0.016, rank-sum test, pre and post saline conditions combined). In contrast, for this set of birds, we did not observe robust effects of social context on syllable sequencing under control saline conditions as has been observed in some previous studies of how social context influences song (*Hampton et al., 2009*; *Toccalino et al., 2016*; *Figure 6—figure supplement 1*; see also Methods). To investigate whether social modulation of song is dependent on cholinergic signaling in HVC, we therefore primarily focused our analysis on pitch, pitch c.v., and

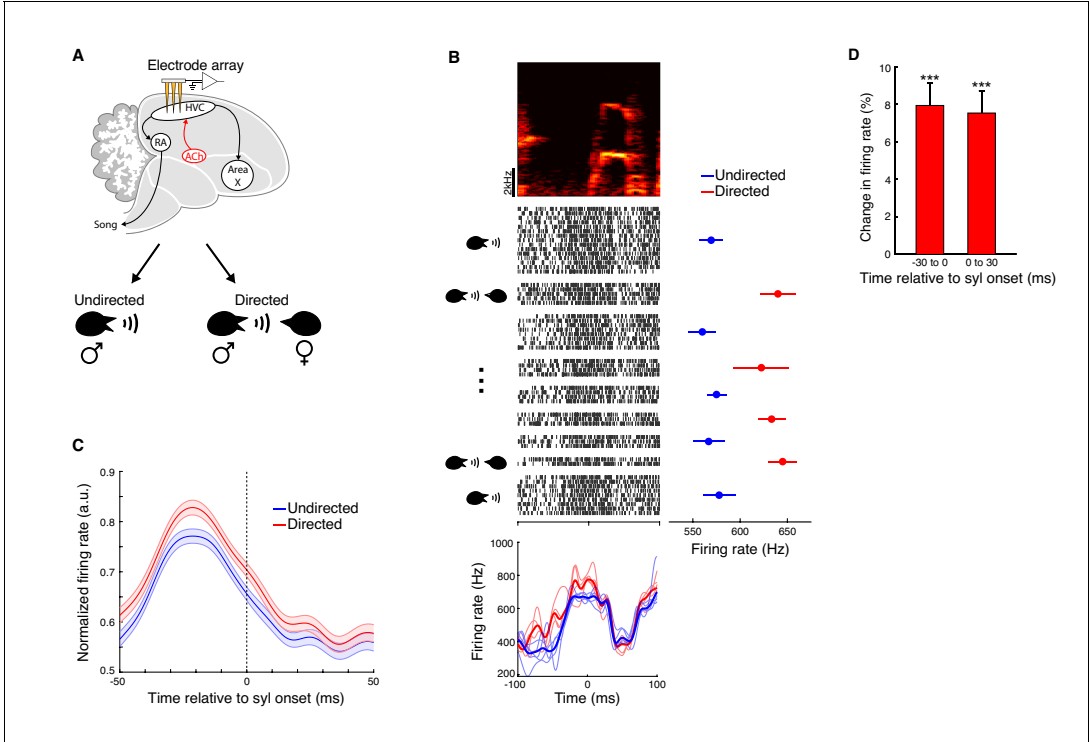

**Figure 5.** HVC activity is modulated by social context. (A) Experiment schematic. Multi-unit activity in HVC was recorded during interleaved female-directed and undirected song sessions. (B) Example multi-unit site (one experiment; activity aligned to the onset of one syllable). Top, spectrogram of the syllable used for alignment. Middle, raster plot of the multi-unit site (activity plotted chronologically from top to bottom; spaces separate blocks of directed or undirected singing). Middle right, mean ± s.e.m. firing rates for each block of singing (firing rates were computed in a 100 ms window centered on the syllable onset). Bottom, rendition-averaged firing rates (smoothed with a 5 ms SD gaussian kernel; bold lines show mean firing rates for all renditions, light lines show mean firing rates for each block of directed or undirected song). (C) Population average firing rates aligned to syllable onsets, for directed and undirected conditions. Prior to averaging across sites/syllables, mean firing rates from directed and undirected renditions were normalized by the maximum of both conditions in a 100 ms window centered on the syllable onset (n = 151 multi-unit sites x syllables, five birds, 10 experiments). (D) Percent change in firing rate during directed song relative to undirected song (mean ± s.e.m. increase in firing rate in −30 to 0 ms window: 7.9 ± 1.2%; 0 to 30 ms window: 7.5 ± 1.2%; directed vs. undirected in −30 to 0 ms window: p=9.7e-11, 0 to 30 ms window: 1.4e-12, firing rate change in −30 to 0 ms window vs. 0 to 30 ms window: p=0.31, two-tailed signed-rank test). ***p<0.001.

The online version of this article includes the following source data and figure supplement(s) for figure 5:

**Source data 1.** Linear mixed effects model analysis for comparing multi-unit firing rates in directed and undirected song.
**Source data 2.** Source data for the summary analysis in figure panels C and D.
**Figure supplement 1.** Example multi-unit sites from each bird demonstrating social modulation of HVC activity.

song tempo—those features that were significantly modulated by social context under control conditions.

We found that blocking muscarinic acetylcholine receptors in HVC with atropine caused an attenuation of social modulation for each of these features, which achieved significance for pitch and pitch c.v., but not song tempo (*Figure 6C–F* and *Figure 6—figure supplement 1*; see also *Figure 6—source data 1* for mixed effects analysis). The reduction in social modulation of pitch and pitch c.v. could in principle reflect an effect of atropine on directed song or on undirected song. We therefore separately analyzed the differences between saline and atropine conditions for each of these features for both directed and undirected song. We found that atropine caused a significant decrease in pitch and increase in pitch c.v. for directed song (relative to saline conditions), but had no effect on these features for undirected song (*Figure 6—figure supplement 2*). Thus, atropine attenuated a process that is specifically engaged during directed song, indicating that increased cholinergic signaling in HVC normally contributes to the increased pitch and reduced pitch variability of directed song.

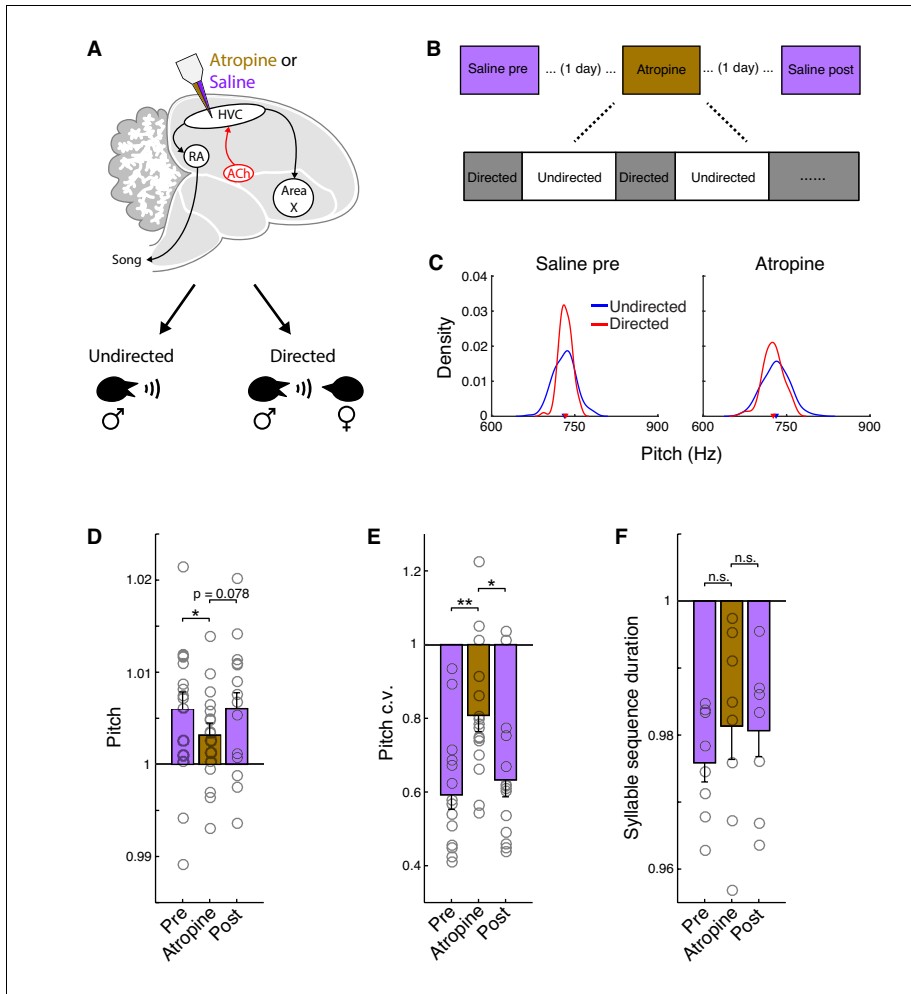

**Figure 6.** Atropine attenuates the social modulation of song. (**A**) Experiment schematic. Atropine or saline was microdialyzed into HVC during interleaved female-directed and undirected song sessions. (**B**) Song was recorded on three separate days for each bird in the following order: Saline pre, Atropine, Saline post, with one day between sessions. See Methods for details. (**C**) Pitch distributions for one example syllable. (**D**) Normalized pitch (directed/undirected) for all syllables (mean ± s.e.m. increase in pitch for Saline pre: 0.59 ± 0.19%, Atropine: 0.32 ± 0.13%, Saline post: 0.60 ± 0.17%; Saline pre vs. Atropine, p=0.046, one-tailed signed-rank test; Saline post vs. Atropine, p=0.078, one-tailed signed-rank test). (**E**) Normalized pitch c.v. (directed/undirected) for all syllables (mean ± s.e.m. reduction in pitch c.v. for Saline pre: 41 ± 3.9%, Atropine: 19 ± 4.4%, Saline post: 37 ± 4.6%; Saline pre vs. Atropine, p=0.0028, one-tailed signed-rank test; Saline post vs. Atropine, p=0.012, one-tailed signed-rank test). For pitch and pitch c.v., n = 16 syllables, six birds, six experiments per condition. (**F**) Normalized syllable sequence duration (directed/undirected) for all syllable sequences (mean ± s.e.m. reduction in sequence duration for Saline pre: 2.4 ± 0.29%, Atropine: 1.9 ± 0.50%, Saline post: 1.9 ± 0.39%; Saline pre vs. Atropine, p=0.098, one-tailed signed-rank test; Saline post vs. Atropine, p=0.47, one-tailed signed-rank test, n = 8 syllable sequences, seven birds, seven experiments per condition). **p<0.01, *p<0.05, n.s., not significant.

The online version of this article includes the following source data and figure supplement(s) for figure 6:

**Source data 1.** Linear mixed effects model analysis of atropine dialysis experiments during directed and undirected song.

**Source data 2.** Source data for the summary analyses in figure panels D–F.

**Figure supplement 1.** Social modulation of syllable durations, gap durations, and song sequencing.

**Figure supplement 2.** Directed but not undirected songs are affected by atropine.

## Discussion

Cholinergic neurons project throughout the forebrain, including motor and sensory cortices, and contribute importantly to global changes in brain activity in aroused behavioral states (*Buzsaki et al., 1988*; *Eckenstein et al., 1988*; *McKinney et al., 1983*; *Metherate et al., 1992*; *Raghanti et al., 2008*). Nonetheless, how cholinergic signaling affects motor cortical activity and behavior remains poorly understood. Here we examined how acetylcholine affects motor behavior in the context of birdsong, leveraging the quantifiable nature of song and the well-defined neural circuitry underlying song production. We found that pharmacological enhancement of cholinergic signaling had an activating effect on HVC and concomitantly increased pitch, amplitude, tempo, and stereotypy of song. These behavioral changes did not require the songbird basal ganglia, indicating that cholinergic enhancement of song vigor occurred via a direct cortical-brainstem pathway. Moreover, the natural increases in pitch and stereotypy of song that are elicited in a courtship context were accompanied by similar increases in neural activity and were attenuated by blockade of cholinergic receptors in HVC. Thus, our findings demonstrate that acetylcholine contributes to enhanced vigor of a motor skill by direct action on cortical premotor circuitry.

### Distributed circuits for the control of motor vigor

Our demonstration that cholinergic invigoration of song does not require participation of basal ganglia circuitry indicates that the neural control of motor vigor is more distributed than has typically been recognized. Prior experimental work and theoretical treatments of motor vigor have primarily focused on basal ganglia circuitry (*Manohar et al., 2015*; *Panigrahi et al., 2015*; *Schmidt et al., 2008*; *Shadmehr et al., 2019*; *Yttri and Dudman, 2016*). While some authors have recognized that cortical and other non-basal ganglia circuitry are likely to contribute to the control of motor vigor (*Dudman and Krakauer, 2016*; *Yttri and Dudman, 2018*), there have been limited experimental tests of this proposal. Our findings indicate that in some situations, motor invigoration can indeed occur independently of basal ganglia circuitry. This separability raises the possibility that cortical and basal ganglia circuitry contribute differentially to the control of movement vigor in distinct contexts. In particular, basal ganglia circuits have been linked to shaping the vigor of motor output in the motivational context of goal-directed behaviors—e.g. movements that are generated to obtain reward, with local dopaminergic signaling playing a key role (*Berke, 2018*; *Dudman and Krakauer, 2016*; *Mazzoni et al., 2007*; *Schmidt et al., 2008*; *Turner and Desmurget, 2010*). In turn, cholinergic modulation of forebrain motor circuitry may especially contribute to invigoration of behaviors that are adaptive in externally-triggered states of heightened physiological arousal, like those that involve escape from danger, prey capture or courtship. Nonetheless, in many naturalistic contexts cortical and basal ganglia circuitry may jointly control movement vigor, with coordination between these pathways mediated by bidirectional feedback between them (*Bosch-Bouju et al., 2013*; *Yttri and Dudman, 2018*). In mammals, projections from cholinergic neurons in the pedunculopontine nucleus to both the cholinergic basal forebrain and dopaminergic neurons in the substantia nigra also link these two pathways (*Bolam et al., 1991*; *Lee and Dan, 2012*), and much of this anatomy is likely to be conserved in songbirds (*Medina and Reiner, 1994*).

### Neural mechanisms underlying the control of motor vigor in songbirds

The persistence of cholinergic effects on song despite the inactivation of basal ganglia circuitry indicates that invigoration of movement can be mediated by direct engagement of cortical projections to brainstem motor nuclei. In the song system, the cortical motor nucleus RA serves as an intermediary between HVC and brainstem motor regions (*Figure 1A*). Our findings indicate that cholinergic invigoration of song can be attributed to altered neural activity within this pathway (from HVC→RA→brainstem). RA also receives input from basal ganglia circuitry via LMAN (*Figure 1A*), so that it is well positioned to integrate both cortical and basal ganglia contributions to song vigor (*Thompson and Johnson, 2007*). Such invigoration likely includes increased activity of RA projection neurons, which has been linked to increased pitch and amplitude via an excitatory influence on the relevant syringeal and respiratory muscles of the vocal apparatus (*Goller and Riede, 2013*; *Sober et al., 2008*; *Srivastava et al., 2015*). Consistent with this model, pharmacological excitation of RA projection neurons (via manipulation of inhibition) produces similar behavioral changes to microdialysis of carbachol into HVC (*Miller et al., 2017*). We therefore predict that the increase in

neural activity in HVC in response to carbachol administration results in net excitation of RA and downstream vocal musculature.

Within HVC, vocal invigoration is likely to be mediated in part by increased activity of the neurons that project directly to RA (the HVC$_{RA}$ neurons), which we hypothesize occurs in conjunction with increased activity of both the basal ganglia projecting HVC$_X$ neurons and local inhibitory interneurons. While our multi-unit recordings cannot directly resolve how these three major HVC cell types are affected by acetylcholine, consideration of our findings in light of prior songbird studies supports the idea that acetylcholine increases both projection neuron and interneuron activity. Multi-unit recordings in HVC are thought to primarily sample from the inhibitory interneurons, due to the fact that this population is considerably more active than the projection neurons during song and awake quiescent periods (*Hahnloser et al., 2002*; *Kozhevnikov and Fee, 2007*; *Liberti et al., 2016*; *Rauske et al., 2003*). Since interneurons provide a strong source of inhibition to projection neurons within HVC (*Kosche et al., 2015*; *Mooney and Prather, 2005*), our finding that acetylcholine increases multi-unit activity suggests that acetylcholine could decrease rather than increase projection neuron activity. However, a model in which acetylcholine suppresses HVC projection neuron activity during song is difficult to reconcile with the cellular effects of acetylcholine measured in acute slice electrophysiology experiments: muscarinic acetylcholine receptor agonists depolarize both classes of projections neurons and hyperpolarize interneurons (*Shea et al., 2010*). Extrapolating from these findings, if acetylcholine applied in vitro hyperpolarizes interneurons, while carbachol applied in vivo causes an increase in their firing rate, then carbachol applied in vivo must recruit a sufficiently strong source of excitatory input to interneurons to overwhelm any suppressive cellular effect that it has. We hypothesize that this excitatory input originates primarily from the local HVC projection neurons (*Kosche et al., 2015*; *Mooney and Prather, 2005*), which are driven to greater activation by the depolarizing effect of acetylcholine. Notably, injections of carbachol into HVC of anesthetized songbirds increase spontaneous activity within RA (*Shea and Margoliash, 2003*), supporting the view that acetylcholine increases the activity of the HVC$_{RA}$ neurons in particular. In sum, our data and prior literature suggest a model in which acetylcholine increases the activity of all major cell classes within HVC in tandem. Such a coordinated increase in activity among the major excitatory and inhibitory cell classes within HVC also occurs normally during the transition from non-singing quiescent periods to active song production (*Kozhevnikov and Fee, 2007*), and is broadly consistent with the tight coupling of excitation and inhibition that has been observed in mammalian cortical regions (*Atallah and Scanziani, 2009*; *Haider et al., 2006*; *Okun and Lampl, 2008*; *Wehr and Zador, 2003*).

## Contributions of HVC to acoustic variability

In the context of song control, our finding that acetylcholine can operate on HVC to reduce pitch variability is somewhat surprising. Most previous studies that have manipulated HVC activity have not reported a reduction in behavioral variability (*Hamaguchi et al., 2016*; *Long and Fee, 2008*; *Zhang et al., 2017*; but see *Isola et al., 2020*). In contrast, lesions and pharmacological inactivation of the AFP output nucleus LMAN reduce pitch variability substantially (*Hampton et al., 2009*; *Kao and Brainard, 2006*; *Kao et al., 2005*; *Stepanek and Doupe, 2010*). The observation that HVC projection neurons exhibit extremely low trial-to-trial variability also might suggest that HVC does not introduce substantial behavioral variability (*Hahnloser et al., 2002*). In contrast, our findings suggest that a significant source of behavioral variability originates within HVC. Conceivably, this variability could be harnessed in the service of reinforcement learning in much the same way that is thought to occur for variability originating from the AFP (*Charlesworth et al., 2012*; *Kojima et al., 2018*).

How might cholinergic modulation of HVC contribute to the observed reduction in behavioral variability? Most simply, acetylcholine could cause decreased rendition-by-rendition variability in HVC$_{RA}$ and/or HVC$_X$ projection neurons that are well-positioned to directly and indirectly control variability in RA. However, since we did not observe reduced neural variability at the multi-unit level, we also consider how changes within HVC could drive reduced behavioral variability even in the absence of reduced variability in projection neuron populations. One possibility is that acetylcholine decorrelates the activity across HVC projection neurons such that their added contributions to motor effectors 'cancel out' (*Darshan et al., 2017*; *Kaufman et al., 2014*; *Sober et al., 2008*). Alternatively, by increasing the firing rates of HVC$_{RA}$ projection neurons (as discussed above), variability within

downstream RA projection neurons could be suppressed through a saturation mechanism, similar to that proposed to account for developmental reduction in vocal variability (*Garst-Orozco et al., 2014*). Additionally, at the network level, increased drive to RA could suppress intrinsic dynamics within RA that amplify external perturbations (originating from LMAN, for example; *Mastrogiuseppe and Ostojic, 2018*; *Rajan et al., 2010*). This in turn could disrupt the correlation of activity within RA that contributes to behavioral variability (*Darshan et al., 2017*; *Sober et al., 2008*).

## A potential role for the cholinergic system in movement disorders

Our finding that arousing or activating stimuli can invigorate movement by mechanisms that are distinct from the action of dopamine in the basal ganglia may explain observations that certain sensory cues and emotional stimuli can have a prokinetic effect on movement disorder patients. Patients with Parkinson's disease or basal ganglia damage engage in fewer volitional movements and exhibit a reduced willingness to exert effort to obtain reward, suggesting that reduced motor vigor in these patients is primarily motivational in nature (*Mazzoni et al., 2007*; *Schmidt et al., 2008*). However, these patients can exhibit 'paradoxical' movements in situations that provoke extreme emotion (*Bonanni et al., 2010*; *Glickstein and Stein, 1991*), and can effectively modulate grip strength when explicitly instructed to do so, even while exhibiting deficits in the modulation of grip strength by reward (*Schmidt et al., 2008*). Similarly, Parkinson's patients can exhibit improvements in movement vigor in response to engaging sensory stimulation (*McIntosh et al., 1997*; *Rubinstein et al., 2002*). Our results indicate that such arousal-dependent motor invigoration in these patients could be enabled by cholinergic modulation of cortical motor circuitry. Observations that dopamine-depleted mice (*Panigrahi et al., 2015*) and patients with Parkinson's disease (*Mazzoni et al., 2007*) can learn to move more vigorously may similarly depend on cholinergic signaling in cortex, consistent with the known role of acetylcholine in motor learning and associated cortical plasticity (*Conner et al., 2003*; *Conner et al., 2005*; *Conner et al., 2010*).

Conversely, some movement disorders that include decreased speed, force, and movement stereotypy may reflect in part disrupted cholinergic signaling in cortical motor regions. Particularly noteworthy in this respect is the slowing of gait, reduced force generation, and loss of verbal fluency that are frequently observed in patients with Alzheimer's disease (*Buchman et al., 2007*; *Ferris and Farlow, 2013*; *Goldman et al., 1999*), which is principally associated with the loss of cholinergic neurons in the basal forebrain and diminished cholinergic innervation of the cortex (*Francis et al., 1999*). Indeed, loss of movement vigor may precede, and be predictive of, subsequent cognitive decline in Alzheimer's and other diseases (*Buchman et al., 2007*). An underappreciated role of cortical cholinergic signaling in the invigoration of movements, as indicated by our findings, may both explain this link, and account for some of the ameliorative effects on movements of pro-cholinergic treatments (*Ferris and Farlow, 2013*).

## Contributions of HVC and the cholinergic system to social modulation of song

Our results demonstrate a previously unknown contribution of HVC to the social modulation of song, contrasting with previous studies that emphasize the role of the AFP (*Hampton et al., 2009*; *Kao and Brainard, 2006*; *Kao et al., 2005*; *Leblois et al., 2010*). While previous neural recordings from $HVC_X$ projection neurons in zebra finches did not reveal conspicuous differences between social contexts (*Woolley et al., 2014*), the other cell classes in HVC were not examined, as was implicitly done here by recording multi-unit activity. However, consistent with our findings, one study of the expression of the immediate early gene (IEG) EGR-1 in HVC of Bengalese finches reported differences between social contexts (*Matheson et al., 2016*). Notably, that study reported greater expression of EGR-1 in the undirected song condition; since EGR-1 expression is often construed as a proxy for neural activity, this finding is potentially at odds with our observation of increased neural activity during directed song. This disparity could reflect differences between the IEG response measured histologically in postmortem tissue and the amount of neural activity during song, arising from the long integration time of the IEG response and/or nonlinearities in the relationship between neural firing rates and IEG expression (*Wang et al., 2019*). Alternatively, this discrepancy could be attributed to differences in the specific neural types that contributed to analysis of neural activity

versus IEG expression levels, or to other experimental variables that may have differed between studies, such as the protocols for eliciting directed song or the regions of HVC that were sampled (*Basista et al., 2014*).

The contribution of the cholinergic system to the social modulation of song is also noteworthy, given that previous work has identified dopaminergic and noradrenergic signaling within song system nuclei as contributing to social modulation of song (*Castelino and Ball, 2005*; *Glaze, 2017*; *Ihle et al., 2015*; *Leblois et al., 2010*; *Sasaki et al., 2006*). Each of these neuromodulators—acetylcholine, dopamine, and norepinephrine—are part of the classical 'ascending arousal system' that responds to activating stimuli and drives changes in internal state (*Lee and Dan, 2012*). Hence, while our results directly demonstrate a strong influence of acetylcholine in HVC on social modulation of song, we expect that multiple neuromodulatory systems are likely to be engaged in a courtship context, orchestrating adaptive changes to song and other courtship behaviors through their collective influence on distributed brain regions.

Beyond the rapid changes in arousal elicited by changes in social context, arousal levels also vary over slower timescales in relation to circadian rhythms, and this variation in arousal may contribute to circadian changes in song and neural activity in song control regions (*Chi and Margoliash, 2001*; *Day et al., 2009*; *Derégnaucourt et al., 2005*; *Glaze and Troyer, 2006*; *Liberti et al., 2016*; *Wood et al., 2013*). Our results suggest that the ascending arousal system, including cholinergic components associated with circadian changes in wakefulness, could contribute to these circadian changes in song structure and neural activity in a similarly distributed manner.

In the context of female-directed song, cholinergic enhancement of motor vigor may function to augment the female's perception of the singer's fitness, or serve a more general function in communication. A corollary of the finding that acetylcholine robustly modulates the acoustic structure of song is that changes to song structure in principle can be decoded to make inferences about the internal state of the singer. That is, greater song vigor in terms of pitch, amplitude, tempo, and stereotypy could be interpreted by a female conspecific as a greater level of arousal and interest on the part of the male. Indeed, behavioral studies have demonstrated that female birds are attentive to the differences between undirected and female-directed song, and that they prefer the latter (*Dunning et al., 2014*; *Woolley and Doupe, 2008*). Similarly, variation in the acoustic structure of song in other settings may serve an adaptive function in communicating levels of arousal, attraction, aggression, or other aspects of internal state. For example, the quality of song in some bird species differs between affiliative interactions in a courtship setting versus territorial interactions between males, or interactions with juveniles (*Chen et al., 2016*; *Trillo and Vehrencamp, 2005*). This parallels the observation for human speech that features such as loudness, pitch, and tempo can be decoded to make inferences about the affective state of the speaker (*Banse and Scherer, 1996*; *Fairbanks and Pronovost, 1938*; *Leinonen et al., 1997*). An intriguing possibility is that for both song and speech, the nuances of acoustic structure that contribute importantly to social communication are driven by the combinatorial influences of neuromodulatory systems that reflect corresponding variations in internal state.

## Materials and methods

### Statistics

Unless noted otherwise, we used nonparametric two-sided tests for comparing two samples: the Wilcoxon rank-sum test for unpaired data and the Wilcoxon signed-rank test for paired data. Details for all statistical tests are included in the figure legends, the main text, or in supplemental tables that accompany the main figures (for linear mixed effects models; see below). For all tests, we rejected the null hypothesis if $p<0.05$. No statistical methods were used to predetermine sample sizes, though our sample sizes are comparable to those used in previous publications (*Sakata and Brainard, 2008*; *Sober et al., 2008*; *Stepanek and Doupe, 2010*). Unless noted otherwise, data collection and analysis were not performed blind to experimental conditions; however, there was minimal opportunity for subjective biases to influence outcomes as exclusion criteria and quantitative analyses were applied uniformly across experimental conditions. Details on randomization of conditions are discussed in the relevant methods section where applicable. A small number of syllables with very low sample sizes were excluded; details are given in the corresponding methods section. For

pitch, tempo, and amplitude measurements, a simple heuristic was used to remove outliers (described in detail below). For amplitude analyses, we excluded a small number of experiments (3/ 75, combined across conditions) in which large (>25%) and sudden changes in amplitude occurred, as these were likely caused by the bird changing its orientation with respect to the recording microphone (described in detail below). Source data are provided for group summary analyses of all of the main figures in associated '.mat' files. MATLAB (MathWorks) code for analyzing these data and generating summary figures is provided in *Source code 1*.

## Mixed effects models

To supplement the primary statistical analysis reported in the main text and figure legends, we conducted an additional statistical analysis using linear mixed effects models to account for structure in the data that may violate the assumption of independence. Specifically, we accounted for correlations between measurements that were sampled from the same bird—both syllables and multi-unit recording sites—by including bird identity as a random effect in a linear mixed effects model. The details of these analyses are provided in supplemental tables that accompany the main figures. These tables are referred to in the text as source data, e.g. *Figure 1—source data 1*.

## Subjects

Data were collected from 25 adult male Bengalese finches (*Lonchura striata domestica*; microdialysis only: n = 15; microdialysis + electrophysiology: n = 5; electrophysiology only: n = 5). All but two birds in the study were bred in the University of California, San Francisco (UCSF) breeding facility. The ages of these birds ranged from 124 to 239 days at the start of experiments. The remaining two birds were obtained from outside sources and had adult-like song and physical characteristics. Adult female Bengalese finches (>120 days old) were used to elicit directed song. During experiments, male birds were individually housed in sound-attenuating chambers (Acoustic Systems) on a 14 hr:10 hr light:dark cycle with food and water provided ad libitum. All procedures were performed in accordance with protocols (#AN170723-02) approved by the UCSF Institutional Animal Care Use Committee.

## Song recording

Audio was recorded with custom Labview software (National Instruments; digitized at 32 kHz) using an omnidirectional lavalier microphone (Countryman), or with a USB interface board (Intan Technologies; digitized at 30 kHz) using a custom-made microphone and pre-amplifier system.

## In vivo microdialysis

Guide cannulae (CMA 7, CMA Microdialysis) were implanted into HVC or both HVC and LMAN using stereotaxic coordinates. For combined electrophysiology/microdialysis experiments, cannulae were implanted unilaterally in the left HVC (n = 5 birds). For all other experiments, cannulae were implanted bilaterally (HVC + LMAN: n = 4 birds; HVC only: n = 11 birds). After birds recovered from surgery, we inserted microdialysis probes into the cannulae (CMA 7; 0.24 mm diameter, 1 mm diffusion membrane, 6 kDa diffusion pore size).

Dialysis probes were connected to fluid pumps through flexible tubing. Outflow was continually monitored throughout the duration of the experiment. In some cases, we observed leakage from the dialysis tubing or diminished flow as indicated by reduced volume of the outflow. These experiments were excluded from summary analyses and dialysis probes were replaced for subsequent experiments. For experiments without combined electrophysiology, solutions were exchanged to either saline (for control experiments) or drug (carbachol, muscimol, etc.) after a two to three-hour baseline period (flow rate held constant at 1–1.5 uL/min.; solutions exchanged at the same time each day across experiments). For experiments with combined electrophysiology, the duration and time of day that solutions were exchanged varied depending on when the bird sang. For animals in which we tested multiple different conditions (e.g., carbachol vs. carbachol + atropine), each condition was repeated a variable number of times on different days in a randomized order. At least one full day of saline-only infusion was interposed between consecutive drug infusion experiments.

In a subset of birds, we conducted a series of pilot experiments to determine effective drug concentrations. For carbachol experiments, we increased the concentration of carbachol until a

significant pitch effect was observed, up to a maximum concentration of 1 mM. In one case, the initial concentration of carbachol (500 µM) caused the bird to call continuously and was reduced on subsequent experiments to 250 µM. For LMAN inactivation experiments, we increased the concentration of muscimol until a significant reduction in pitch c.v. was observed, or to the highest level that did not interfere with singing. For combined microdialysis and electrophysiology experiments, no pilot experiments were conducted, and all experiments were conducted with 1 mM carbachol. Pilot experiments were not included in summary analyses. The final concentration of drugs used in this study is as follows. Carbachol (Santa Cruz Biotechnology): 250 µM-1mM; muscimol (Tocris): 250 µM-1mM; mecamylamine hydrochloride (Sigma, abbreviated MEC): 400 µM; atropine sulfate (Sigma, abbreviated Atrp): 500 µM-2mM; methyllycaconitine citrate salt (Sigma, abbreviated MLA): 100 µM. In cases where more than one concentration of antagonist was used within a given animal, we included only data from experiments with the highest concentrations. All drugs were diluted in saline.

While we did not directly confirm the extent of drug diffusion, previous studies in songbirds that used a similar dialysis procedure estimated that the radius of maximal spread of biotinylated muscimol is <= 1 mm (*Warren et al., 2011*). In contrast, the boundaries of the nearest major song control nuclei (NIf and RA) are ~2–3 mm away from our probes, implying that diffusion of drug to these regions is minimal.

## In vivo electrophysiology

For experiments in which we combined electrophysiology with microdialysis (n = 5 birds), extracellular recordings from HVC were obtained with custom-designed tungsten electrode arrays (MicroProbes, 10 electrodes, 6MOhm impedances, n = 1 bird), multi-site silicon electrode arrays (NeuroNexus, A4 × 4–3 mm-50-125-413-H16_21 mm, signal acquired from 12 out of 16 sites, n = 1 bird), or tungsten electrode arrays assembled in-house (FHC or MicroProbes, 5–11 electrodes per array, impedances ranging from 0.5MOhm to 10MOhm, n = 3 birds). Electrode arrays were positioned using a custom manual microdrive.

For experiments in which we monitored HVC activity during both directed and undirected song (n = 5 birds), extracellular recordings were obtained with tungsten electrode arrays assembled in-house (MicroProbes, 4–10 electrodes per array, impedances ranging from 0.5MOhm to 6MOhm). Electrode arrays were positioned using a custom manual microdrive, or remotely using a custom motorized microdrive (Faulhaber motor, n = 1 bird).

Neural data were amplified, band-pass filtered (1–7500 Hz), and digitized (30 kHz) with a commercially available head-mounted amplifier board (Intan Technologies, RHD2132 16-channel amplifier board, part #C3334) or a custom amplifier board designed in-house to reduce weight, made with the RHD2132 amplifier chip (Intan Technologies). Neural and audio data were registered with a USB interface board (RHD2000, Intan Technologies).

## Female-directed song

For experiments in which we microdialyzed atropine into HVC while manipulating social context, we collected female-directed song and interleaved undirected song on three separate days, with sessions separated by one day (day 1: saline, day 3: atropine, day 5: saline; see *Figure 6B*). The final saline day was included to ensure that any attenuation of social modulation of song by atropine could not be attributed to habituation of courtship behavior over time or repeated exposure to females (*Hampton et al., 2009*; *Toccalino et al., 2016*). For each session, females were presented 30 or 40 min apart in a cage placed next to the male's cage (spacing between sessions was constant for a given bird), for a total of 2 min for each presentation. For a given male bird, the same sequence of females was presented in the same order across sessions, since some aspects of courtship behavior can depend on female identity (*Heinig et al., 2014*). The presence of a courtship dance was used to confirm that males sang female-directed song (puffing of feathers, orientation toward female, and hopping from side-to-side). For experiments in which we recorded neural activity in HVC while manipulating social context, females were presented 15–20 min apart in a cage placed next to the male's cage, or were introduced directly into the male's cage.

## Analysis of song features

### Definition of analysis windows

To quantify the behavioral effects of carbachol, we analyzed songs recorded in a two-hour window beginning one hour after the estimated time of drug delivery into the brain (drug analysis window). This same choice of analysis window was used for all animals and experiments, and for all reported behavioral features. We selected this time window primarily to ensure that carbachol would have time to diffuse throughout HVC. However, this time window also avoids transient behavioral effects that were observed following carbachol infusion. Such transient effects typically included a gradual onset of reported behavioral changes, consistent with prior studies indicating that it can take tens of minutes for dialyzed drugs to diffuse throughout targeted structures. We also occasionally observed more idiosyncratic patterns of behavioral change following onset of drug infusion (see *Figure 3B*, for example), but because these were inconsistent across experiments and transient, we did not analyze them further, instead focusing on the relatively robust, consistent, and stable effects that persisted throughout our analysis window. Baseline measurements were obtained from songs recorded in a one to two-hour window immediately prior to the exchange of dialysis solutions (baseline analysis window).

For experiments in which we probed the requirement of LMAN for the behavioral effects of carbachol, we estimated the onset of drug effects for the carbachol and muscimol conditions by visual inspection of the raw pitch time course from select syllables. We then defined the drug analysis window as a one-hour window beginning at the maximum of the estimated carbachol and muscimol onsets (held constant for a given animal across conditions). This procedure ensured that both drugs would be active during the combined carbachol + muscimol condition. The baseline analysis window was defined as described above. For social context experiments, we analyzed all recorded directed songs and a random subset of interleaved undirected songs.

### Outlier and experiment inclusion criteria

Unless otherwise noted, we excluded pitch, amplitude, and tempo measurements that exceeded four times the median absolute deviation from the median, repeating this process three times. This procedure functions as a simple heuristic for culling erroneous measurements resulting from incorrect segmentation of the amplitude envelope. For a given experiment, syllables, syllable sequences, or branch points with fewer than 15 recorded renditions in any condition (e.g. drug or baseline periods, undirected or directed songs) were not included in summary analyses. This criterion was applied after outlier removal.

### Pitch and pitch variability

Analysis of fundamental frequency (pitch) was carried out on a random subset of 'harmonic stack' syllables (i.e. syllables with clear harmonic structure that was relatively constant over the duration of the syllable). We focused on such harmonic stacks, as in many previous studies (*Kao and Brainard, 2006*; *Kao et al., 2005*; *Kojima et al., 2018*; *Stepanek and Doupe, 2010*), because it is difficult to consistently quantify the fundamental frequency of syllables that have entropic spectral structure ('noisy syllables') or frequency modulated components ('sweeps'). For sweeps in particular, the value of fundamental frequency changes rapidly over the course of the syllable so that the value measured for a particular rendition is sensitive to where within the syllable the measurement is made. Changes in song tempo (and syllable duration) exacerbate this problem, and can confound pitch measurements when comparing conditions across which song tempo varies. For quantitative pitch measurements, we therefore used harmonic stack syllables, for which we confirmed that estimates of pitch are robust to variation in the location within the syllable at which measurements are made.

To quantify the pitch of a given syllable rendition, raw audio data were bandpass filtered between 500 and 10,000 Hz, and a spectrogram was computed using a gaussian-windowed (SD = 1–3 ms) short-time Fourier transform (window size = 1024 samples; overlap = 1020 samples). A pitch contour was calculated from the spectrogram by finding the maximum power in a small frequency range around the first harmonic in each time bin, followed by parabolic interpolation of the resulting time series. The pitch of the syllable was then determined by averaging the pitch over a fixed portion of the syllable (relative to syllable onset) that had a constant frequency component. The coefficient of

variation of pitch (pitch c.v.) was computed as the across-rendition standard deviation divided by the mean.

For social context experiments, the accuracy of each pitch calculation was confirmed by visual inspection of the pitch contour overlaid on the syllable spectrogram, and inaccurate pitch calculations (due to incorrect segmentation, for example) were excluded from summary analyses. Exclusions were performed blind to social context condition (female-directed vs. undirected). No additional outlier removal was performed.

## Song tempo

We calculated the duration of one or two stereotyped syllable sequences from each bird's repertoire (i.e. sequences without sequence variability; range: 2–4 syllables). Sequence duration was determined from the onset of the first syllable in the sequence to the onset of the last syllable in the sequence. Syllable onsets were determined by an amplitude threshold and were used for tempo measurements because they were more sharply defined than syllable offsets. We also measured the durations of syllables and gaps (using an amplitude threshold). Only gaps that occurred in stereotyped syllable sequences were included in summary analyses.

## Amplitude

Amplitude for a given syllable was calculated by averaging the smoothed amplitude envelope over the middle 80% of the syllable. Amplitude envelopes were calculated by bandpass filtering the raw audio signal between 500 and 10,000 Hz (80th order linear-phase FIR filter), computing the root-mean-square, and smoothing with a sliding 2.5 ms rectangular window. In a small number of experiments, we observed large (>25%) and sudden changes in amplitude that were likely caused by the bird changing its orientation with respect to the recording microphone. We excluded these experiments if the mean amplitude for each syllable changed by more than 25% (in the same direction) in the drug period relative to the baseline period (3/75 experiments excluded, combined across conditions).

For analyses in which we measured changes in amplitude for different types of syllables (*Figure 1—figure supplement 1*), we defined the following four types of syllables: 'harmonic stacks' and 'sweeps' were defined by—respectively—constant or frequency-modulated (FM) harmonics for the majority of the syllable's duration; 'complex' syllables contained both a FM sweep and either a constant harmonic component or a high-entropy component; and 'noisy' syllables did not fit into these other three categories and were typically characterized by high spectral entropy. To increase the sample size of non-harmonic syllables, we labelled an additional n = 21 syllables in a subset of experiments; these syllables are not included in the main amplitude group summary analysis (*Figure 1I*).

## Repeated syllables

Syllables that repeated a variable number of times were classified as repeated syllables, with the following exception: syllables that repeated only once or twice were considered as branch points (see below). We also did not consider high entropy syllables that sometimes separate motifs in Bengalese finch song as repeated syllables, since these are difficult to distinguish from introductory notes. Syllables separated by a gap of more than 200 ms were not considered a part of the same repeat sequence. The primary cohort of animals used in this study had few repeated syllables and so we included an additional cohort of animals that did not have paired saline control experiments (n = 4 additional animals). Prior to statistical comparisons and calculation of normalized repeat length, we pooled syllable renditions across experiments. Repeat length c.v. was calculated from the pooled repeat length distributions from the baseline period of all carbachol experiments. For the social context experiments, we assessed whether atropine attenuated the social modulation of two repeated syllables that exhibited a significant increase in repeat length during directed song relative to undirected song (directed vs. undirected, p<0.05 for both saline pre and saline post, one-tailed rank-sum test; n = 2 birds, two experiments per condition). Three other repeated syllables did not exhibit significant social modulation and therefore were not analyzed for attenuation by atropine (directed vs. undirected, p>0.05 for saline pre and saline post, one-tailed rank-sum test).

## Branch points

A syllable that transitions probabilistically to two or more syllables is a branch point (specifically, a divergent branch point). Similar to previous studies (*Zhang et al., 2017*), we treated sequences of repeated syllables as a single song element. Syllables separated by a gap of more than 200 ms were not included in the calculation of transition probabilities.

To determine if transition probabilities at a given branch point were significantly different in the baseline and drug windows, we employed a generalized likelihood ratio test for homogeneity of transition probabilities. Specifically, we tested the null hypothesis $H_0$: $p_i = q_i$ for all $n$ possible transitions; where $p_i$ denotes the probability of transition $i$ in the baseline period, and $q_i$ denotes the probability of transition $i$ in the drug period. The test statistic is the likelihood ratio $L(M_{sub})/L(M_{full})$, where $M_{full}$ denotes two independent and unconstrained multinomial distributions with parameters estimated separately for the baseline and drug periods, and $M_{sub}$ denotes a single multinomial distribution with parameters estimated from the combined baseline and drug periods. Intuitively, this ratio captures the extent to which a single multinomial model is a better descriptor of the data than two separate multinomial models split by baseline and drug periods, thereby adjudicating the hypothesis that transition probabilities have changed.

Systematic differences in sample sizes between carbachol and saline experiments could confound the interpretation that carbachol affects sequencing, as assessed by analyzing the proportion of branch points with significant changes in transition probabilities. However, the number of transitions from the combined baseline and drug periods did not differ between carbachol and saline experiments (p=0.89, two-tailed signed-rank test, n = 15 branch points, seven birds).

For a given branch point, the magnitude of change in transition probability was calculated as the summed change in transition probability for the first $n$-1 of $n$ possible transition types. Prior to calculating this statistic, we pooled data across experiments. Transition entropy at variable transitions in song was calculated as $\sum -p_i \cdot \log_2(p_i)$, where the sum ranges over the $n$ possible transitions. For the social context experiments, we did not observe a significant decrease in transition entropy for directed song relative to undirected song, rendering a comparison to atropine irrelevant (mean ± s.e.m. decrease in transition entropy for Saline pre: 3.5 ± 5.6%, Saline post: 0.75 ± 17%; undirected vs. directed for Saline pre: p=0.41, Saline post: p=0.15, one-tailed signed-rank test).

## Neural analyses

### Spike sorting

Multi-unit activity was extracted using Wave clus (*Quiroga et al., 2004*). Briefly, raw voltage traces from all recorded song files were concatenated and bandpass filtered between 300 and 4000 Hz. Events greater than 3.5 and below 50 times the estimated noise level in the negative direction were considered spikes, with a minimum refractory period between events of 0.2 ms. Wave clus estimates noise as the median absolute deviation of the filtered voltage trace, divided by 0.6745 (*Donoho and Johnstone, 1994*; *Quiroga et al., 2004*). This noise estimator mitigates the upward bias that would be introduced by instead using the standard deviation of the signal, since the signal contains a small fraction of large events (corresponding to spikes).

### Signal-to-noise ratio of multi-unit data

We estimated the signal-to-noise ratio (SNR) of our multi-unit recording sites as the mean peak height of all detected events, divided by the estimated noise level as defined above for spike sorting. The SNR was calculated separately for all recording sites and syllables. Since the estimated noise level for data in the vicinity of a given syllable could differ from that obtained from the entire recording (as was done for spike detection), a small fraction of our site/syllables had SNRs below the 3.5x spike detection threshold.

### Analysis of firing rates and neural variability

Rendition-averaged firing rates before and after carbachol/saline, and for directed and undirected song, were calculated by aligning spike trains to syllable onsets and convolving with a 5 ms SD gaussian kernel. Firing rate differences elicited by carbachol or directed song were calculated by averaging smoothed firing rates over three separate time windows: a 100 ms window centered on syllable onsets, a 30 ms window just prior to syllable onsets, and a 30 ms window just after syllable

onsets. The Fano factor was calculated as the across-rendition spike count variance divided by the mean spike count in these same time windows. For all analyses, multi-unit site/syllable pairs with fewer than 10 renditions or firing rates less than 50 Hz in either the baseline/undirected or drug/directed periods were excluded. The minimum firing rate criterion was applied to a 100 ms window centered at the onset of the syllable.

## Localization of microdialysis probes and recording electrodes

We collected post-mortem histology at the conclusion of experiments to confirm the placement of microdialysis probes and recording electrodes. HVC was visualized by fluorescent staining for Parvalbumin (Swant Cat# 235, RRID:AB_10000343; monoclonal ab raised in mice, 1:10000); LMAN was visualized by fluorescent staining for calcitonin gene-related peptide (Sigma-Aldrich Cat# C8198, RRID:AB_259091; polyclonal ab raised in rabbits, 1:5000 to 1:10000). The location of microdialysis probes was indicated by tissue damage within or adjacent to HVC or LMAN. Placement of recording electrodes was confirmed by tracks left by the electrodes and/or small electrolytic marker lesions. Dialysis probe placement could not be confirmed in 4/20 birds that died without being perfused for histology. However, we observed behavioral effects in these animals in response to drug microdialysis (i.e. increased pitch/amplitude/tempo and reduced pitch c.v. for microdialysis of carbachol into HVC, and reduced pitch c.v. for microdialysis of muscimol into LMAN), providing an independent confirmation of correct probe placement.

## Acknowledgements

We thank Josh Berke, Alla Karpova, Lucas Tian, and members of the Brainard lab for helpful discussion and comments on the manuscript. We also thank Hamish Mehaffey for technical assistance and for designing the lightweight headstages used for extracellular recordings. This work was supported by the Howard Hughes Medical Institute.

## Additional information

### Funding

| Funder | Author |
| --- | --- |
| Howard Hughes Medical Institute | Michael S Brainard |

The funders had no role in study design, data collection and interpretation, or the decision to submit the work for publication.

### Author contributions

Paul I Jaffe, Conceptualization, Software, Formal analysis, Investigation, Methodology, Writing - original draft, Writing - review and editing; Michael S Brainard, Conceptualization, Resources, Methodology, Writing - review and editing

### Author ORCIDs

Paul I Jaffe ⓘ https://orcid.org/0000-0003-0680-3923
Michael S Brainard ⓘ https://orcid.org/0000-0002-9425-9907

### Ethics

Animal experimentation: All procedures were performed in accordance with protocols (#AN170723-02) approved by the UCSF Institutional Animal Care Use Committee.

### Decision letter and Author response

Decision letter https://doi.org/10.7554/eLife.53288.sa1
Author response https://doi.org/10.7554/eLife.53288.sa2

## Additional files

### Supplementary files
• Source code 1. Source code for generating figures.

• Transparent reporting form

### Data availability
Source data are provided for group summary analyses of all of the main figures in associated ".mat" files: Figures 1D, 1E, 1G, and 1I; Figures 2C and 2F; Figures 3C-F; Figures 4C-F; Figures 5C and 5D; and Figures 6D-F. MATLAB code for analyzing these data and generating summary figures is provided as Source code 1.

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
