## [Decision Letter]

**Acceptance summary:**

Although the cholinergic system is well known to modulate sensory processing, its effects on motor systems are less well studied. The reviewers were enthusiastic about this paper's demonstration of cholinergic modulation of motor behavior within the birdsong premotor nucleus HVC. Several aspects of the cholinergic modulation of motor outputs described here have typically been investigated within the context of the dopaminergic system. In particular, the findings suggest that cholinergic basal forebrain inputs to HVC may contribute to the well-described modulation of song in the presence of a female bird (directed song). Together, these findings raise intriguing possibilities about potential contributions of cholinergic signaling to modulation of motor outputs, across systems.

**Decision letter after peer review:**

Thank you for submitting your article "Acetylcholine acts on songbird premotor circuitry to invigorate vocal output" for consideration by *eLife*. Your article has been reviewed by Ronald Calabrese as the Senior Editor, a Reviewing Editor, and three reviewers. The following individuals involved in review of your submission have agreed to reveal their identity: Stephen D Shea (Reviewer #1); Joshua Tate Dudman (Reviewer #2); Sarah C Woolley (Reviewer #3).

The reviewers have discussed the reviews with one another and the Reviewing Editor has drafted this decision to help you prepare a revised submission.

Summary:

The cholinergic system is well known to modulate sensory processing. The paper by Jaffe and colleagues describes cholinergic modulation of motor behavior within the birdsong premotor nucleus HVC. Their findings further suggest that cholinergic basal forebrain inputs to HVC may contribute to the well-described modulation of song in the presence of a female bird (directed song). Several aspects of the cholinergic modulation of motor outputs that they describe have typically been investigated within the context of the dopaminergic system. Together, these findings raise intriguing possibilities about potential contributions of cholinergic signaling to modulation of motor outputs, across systems.

All reviewers were in agreement that this work is well-executed and interesting. There was also broad agreement on the suggestions for improvement, which are summarized here.

Essential revisions:

1) Several reviewers commented on the limits of multiunit recordings in general and more specifically in HVC because of the disparity in firing rates among cell types. Please provide better characterization of the nature of the recordings and clearer expression of some of the caveats associated with this approach. Discussing the results in light of what we know about the cell types and connectivity in the local network (interneurons vs. RA-projecting neurons, etc.) would also be helpful.

2) All of the reviewers would like to see a more thorough treatment of the effects on syllable sequencing, for both the carbachol and atropine experiments, particularly with regard to branch points. Also of interest is HVC activity at these branch points, if available.

3) Please provide a finer grain analysis of the durations of individual syllables and gaps in directed vs. undirected song as opposed to the overall length of predominant motifs.

We expect that these points, as well as the reviewers' individual concerns (appended below), can be addressed without new experiments, and with a moderate amount of further analysis.

*Reviewer #1:*

In this paper, "Acetylcholine acts on songbird premotor circuitry to invigorate vocal output," Jaffe et al. report several pieces of evidence that suggest that a previously described input to the song premotor nucleus HVC from the cholinergic basal forebrain regulates song motor behavior. The experiments are skillfully and rigorously conducted, and there are several novel aspects to the conclusions. First, as the authors correctly note, cholinergic release in the cortex or other forebrain targets are long known to modulate sensory activity. However, the contribution of these circuits to motor performance are more poorly understood. Second, it is also long been known that there are significant differences in the structure of songs performed in the presence of a female ('directed song') as compared to songs performed while the male is alone ('undirected song'). Over the past 15 years or so, a number of features of the circuitry underlying this phenomenon have been discovered. For the most part, these studies have focused on the dopamine system and the so-called 'anterior forebrain pathway,' a basal ganglia-thamalocortical loop that is been implicated in changes to song in both juveniles and adults. The novelty of this paper primarily lies in its identification of cholinergic pathways to HVC as a separate neurochemical arousal pathway that makes an overlapping contribution to social modulation of song via a distinct song system target. With this study, Jaffe et al. expand our understanding of the mechanisms of contextual regulation of vocal behavior, and they reveal an unappreciated function of the cholinergic system, which is understudied in songbirds. I have a number of more specific comments listed below.

1) Acetylcholine is of course involved in arousal, and the authors have good evidence that this pathway is activated during the arousal that accompanies directed song. Nevertheless, the cholinergic system is also a major participant in circadian patterns of arousal and alertness. In light of the sizable body of data that show circadian variability in song behavior and neural activity in the song system, some speculation in the discussion about how their data might interface with that line of work seems warranted.

2) I recommend the authors add citations of these relevant papers to their Introduction: Neuron 46:173; Cerebral Cortex 20:2739.

3) I understand why the authors limited their pitch analysis to syllables with defined harmonic structure that were also constant in pitch, but the authors might consider analyzing a subset of different syllables that don't share these attributes to see if an increase in pitch is global to song or if noisier or modulated syllables are affected differently. For example, syllables that contain substantial spectrotemporal modulation may show an increase in the span of that spectrotemporal modulation rather than a uniform upward shift in pitch.

4) I'm curious why the authors did not separately analyze the duration of each syllable and silence between them, instead choosing to analyze the overall duration of common groups of syllables. As far as I know, this is not the approach taken by past studies of contextual modulation of song behavior.

5) For me, one of the most interesting results was the experiment depicted Supplementary Figure 1. The authors' finding that a manipulation of HVC alone affected song sequencing is quite noteworthy. As far as I understand, the neural mechanisms of song sequencing, particularly in species with significant sequence variability like Bengalese finches, remain controversial. This result seems to support the idea that song sequencing is entirely contained within the HVC circuit. Therefore, I'm not sure these data deserve to be relegated to a supplementary figure. That said, if the authors were to move it to a main figure, they should probably do a more comprehensive analysis of sequence changes. I think here the focus on the most common sequences, which like in point #1 above, may be affected differently than rarer sequences.

6) I noticed something in Figure 2B that may be interesting if it is consistently observed. In these examples, the carbachol experiment in which LMAN is intact shows a larger pitch shift in earlier songs that relaxes partially in later songs. This is not seen in the example they show in which LMAN has been inactivated. Was this something they observed consistently? Because it might suggest that after many songs performed in the presence of carbachol, the bird begins to correct or compensate for the pitch shift, and that this may be LMAN-dependent.

7) I'm not sure that the authors assertion that HVC firing rates are specifically elevated just before the syllable onset is very well justified by the data. Although there is a difference in Figure 3E between firing rates just before and just after the start of the syllable, there is still a large increase in both in the presence of carbachol. Also, why did they not perform this temporally limited analysis on the data in Figure 4?

8) There are at least three major classes of neurons in HVC based on their projection targets: RA-projecting, X-projecting, and interneurons. The projection neurons fire much less frequently than the interneurons, so their multiunit recording data is almost certainly overwhelmingly dominated by the activity of interneurons. This is an important point that, in my opinion, they have to address because (a) the activity the authors see likely doesn't reflect much of the activity of the pathway they think is mediating the effects of carbachol, and (b) the interneurons are GABAergic and probably target RA-projecting neurons with synaptic inhibition. This leads to potentially a different interpretation of the changes in HVC firing rate that they see in carbachol infusion and directed song.

*Reviewer #2:*

Here the authors describe a set of experiments to assess the direct contributions of cholinergic modulation of activity in a premotor nucleus on the execution of bird song – specifically the vigor with which the song is performed. This touches upon longstanding ideas that cholinergic modulation of cortical circuits emanating from the basal forebrain play a critical role in modulating activity. This modulation has often been associated with generalized increases in processing such as attention, arousal, sensory gain etc. It is relatively less well studied what contribution such modulation may play on the details with which actions are executed. The latter has been more closely associated with basal ganglia function in the literature. Moreover, as far as I could tell this question is largely unexplored specifically in the bird. I found the experiments well done and compelling. The authors provide clear evidence that muscarinic modulation of HVC activity can alter the vigor of song, that this parallels and potentially mediates some portion of the social modulation of song performance, and occurs as an independent, additive modulation with modulation by basal ganglia circuits. I think the paper merits publication. Below I detail some points that I think could be refined in a revised version of the manuscript and one experimental/analysis question that I think would benefit the paper if addressed.

A comment on analysis:

"Thus, the increased pitch stereotypy caused by carbachol cannot be explained by reduced neural variability in HVC."

The authors do not really return to this point in the Discussion section, but I was interested to understand why/how this dissociation might be relevant to theories about syllable production and variability in pitch that is exploited in many reinforcement paradigms.

The transition probability effect at branch syllables was not, as far as I could find, addressed in light of recordings from HVC. It seemed however that since HVC activity sequences are thought to determine transition probabilities this would be quite interesting to explore. Was it not possible to observe HVC activity specifically around branch syllables for sampling reasons? Or were there other reasons this was not explored in more detail? Does social modulation also modulate transition probabilities?

A general point:

At least in mammals ACh also acts within BG via intrinsic cholinergic neurons in striatum and globus pallidus as well as projections from midbrain (PPN for example) on to midbrain dopamine neurons. In the design of the current experiments the coordination of ACh signaling mediated by these pathways might be lost. It would be useful to discuss the point that while the Ach-mediated vigor mechanisms can be dissociated from putative dopamine-dependent mechanisms using local infusion of carbachol, the two mechanisms may in fact be coordinated in other conditions. Perhaps consistent with this in Figure 5 it would appear that HVC-infused atropine mediates only part of the social modulation of song consistent with possibility that mAChR function in other (possibly BG-related) brain areas could contribute to further effects. This might make additional infusions of atropine in the context of directed song interesting to explore whether cholinergic modulation more broadly than in HVC is necessary to account for the full extent of social modulation.

Comments on text/interpretation:

"the extent to which acetylcholine contributes to enhanced motor vigor observed in aroused behavioral states remains unknown."

Maybe modify this point a bit. There is a certainly a strong association between cholinergic activity and aroused behavioral states observed in mammals (Jones, 2004) and there has long been data on how mAChR antagonists effect aroused behavioral responses (such as conditioned responses, e.g. Longo, 1966).

"Moreover, for each feature, the changes elicited by combined carbachol + LMAN inactivation were not significantly different from the sum of the individual effects of carbachol and LMAN inactivation (pitch: p = 0.77, pitch c.v.: p = 0.85, tempo: p = 0.22, amplitude: p = 0.71, signed-rank test). These results indicate that increased cholinergic tone in HVC can modulate song via primary motor circuitry independently of input from the songbird basal ganglia."

I thought this result was a particularly key demonstration that these two pathways appear to linearly sum. I would note that a linear sum of BG-independent and BG-dependent mechanisms has been proposed previously (Yttri and Dudman, 2018) – although this provides the clearest direct evidence for ~linear combination of effects to date.

“Prior work has identified basal ganglia circuitry as an important locus for the control of motor vigor (Panigrahi et al., 2015; Schmidt et al., 2008; Yttri and Dudman, 2016). In principle, cortical and basal ganglia circuitry could jointly control movement vigor, with coordination between these pathways mediated by bidirectional feedback between them (Bosch-Bouju et al., 2013). However, our findings indicate that in some situations, modulation of motor vigor can occur independently of basal ganglia circuitry.”

The Discussion section as written suggests that previous treatments have suggested that basal ganglia (BG) might be an exclusive determinant of motor vigor. However, I would just note that this was explicitly presented as an argument for BG-independent pathways for control of movement vigor previously – as noted in Dudman and Krakauer (2016) cited in this manuscript: "It should be stressed that just because the basal ganglia can influence vigor this does not imply the converse: that vigor parameters are always under the obligate control of the basal ganglia." This is also explicit in normal treatments of this model of vigor in the equations in Yttri and Dudman (2018) in which BG-independent component of vigor sums (is independent of) the BG contribution.

Nonetheless, I think the authors make a very important point here that an explicit modulatory influence that does function in cortex independent of BG to control vigor is a very valuable addition/demonstration. The authors might be interested to consider other phenomena that are similarly hard to reconcile with an exclusive reinforcement mechanism like verbally-instructed changes in movement vigor in normal subjects ("Move faster!"), or the learned changes in vigor present that persist in dopamine depleted animals (Panigrahi et al., 2015) and patients (Mazzoni et al., 2007; Baraduc et al., 2013), or as authors do discuss paradoxical kinesia. (These were the observations that led to the proposal ofBG-independent circuits for motor vigor in previous reviews). Moreover, similar to the bird circuitry highlighted in the discussion of this manuscript, in the mammal BG-output also converges on subcortical targets of descending motor cortical projections making these parallels perhaps even closer.

*Reviewer #3:*

In this manuscript, Jaffe and Brainard investigate the role of acetylcholine in motor invigoration. In general, I found this to be a thorough manuscript addressing an interesting question of how acetylcholine modulates motor output. That said, my enthusiasm is tempered by a handful of issues regarding the approach and aspects of the data that were not included or not sufficiently described.

1) Historically, HVC has been thought to be important for tempo and sequencing while the AFP contributes to syllable structure. This makes Bengalese finches an interesting model to study HVC because they have greater sequence variability than zebra finches (a focus that Brainard has taken advantage of in the past). In particular, his lab has found that during directed singing, sequence entropy decreases and syllable repeats increase and this has not been found to be affected by lesions of LMAN, thus hinting that HVC may be significant in this modulation. Here, they report that Ach manipulation in HVC affects sequence transitions (though there is little detail on this change, including whether there is a change in entropy) and also increases repeats (similar to the effect of directed singing). However, there is no further discussion of sequence or repeats through the rest of the paper. I would like to see the effects on sequence and repeats throughout. It would be especially interesting to know whether atropine affects the decrease in entropy and increase in repeats previously reported for directed singing. Presumably, this would not require additional experiments as these data would be available in the data already collected.

2) I'd like more raw data and interpretation of the multi-unit recordings. It's unclear what these sites look like (how multi-unit is multi-unit?). Moreover, I find the rasters to be a poor way to convey the effects. In the one in Figure 3, there is so much black that you notice the decrease in white-space more than the increase in black, which gives the impression of a decrease rather than an increase. In Figure 4 I can't see the difference in the rasters. For interpretation, is the idea that there are just more neurons firing during directed singing, or that there is an increase in the rate of individual neurons, or both? Work using IEG expression in Bengalese finches indicates that there are more EGR1 expressing cells during undirected singing that directed singing, which would appear to be at odds with the current multi-unit result, but this is not discussed.

3) There are a number of instances in which there are many syllables and/or recordings sites in a single bird but there is no indication that bird has been included as a random variable in the statistical models. It is actually quite important to include bird ID as a variable in the model to account for the fact that many measurements have been made in the same individual (or to indicate more explicitly if it has already been included).

---

## [Author Response]

Essential revisions:1) Several reviewers commented on the limits of multiunit recordings in general and more specifically in HVC because of the disparity in firing rates among cell types. Please provide better characterization of the nature of the recordings and clearer expression of some of the caveats associated with this approach. Discussing the results in light of what we know about the cell types and connectivity in the local network (interneurons vs. RA-projecting neurons, etc.) would also be helpful.

The revised manuscript now includes substantial additional characterization of our multi-unit recordings, including examples of raw data and a quantification of the signal-to-noise ratio and firing rates for each recording site (Figure 4—figure supplement 1). Additionally, our revision now contains a more extended and nuanced discussion of how the different cell types within HVC may be affected by acetylcholine and contribute to the observed behavioral effects, incorporating previous literature pertaining to the connectivity within HVC and cellular effects of acetylcholine as assessed in acute slice electrophysiology experiments (Shea et al., 2010; see subsection “Neural mechanisms underlying the control of motor vigor in songbirds”). More detail on the particulars of these revisions is provided in our responses to Reviewer #1, point #8 and Reviewer #3, point #2.

2) All of the reviewers would like to see a more thorough treatment of the effects on syllable sequencing, for both the carbachol and atropine experiments, particularly with regard to branch points. Also, of interest is HVC activity at these branch points, if available.

As requested, we now provide a more thorough characterization of the syllable sequencing effects for both the carbachol and atropine experiments. Our new Figure 2 and associated text presents data showing that carbachol has a significant effect on sequence probabilities for both branch points (Figures 2A-C) and repeated syllables (Figure 2D-F). Our new Figure 6—figure supplement 1 and associated text presents new analysis addressing how atropine influences sequencing for the limited set of branch points and repeated syllables present in the relevant dataset (i.e. songs from the subset of birds singing both directed and undirected song with or without atropine dialyzed into HVC). For this dataset, the effects of directed song on sequencing were weak even under control saline conditions (as has sometimes been observed previously; see for example Hampton et al., 2009 Figure 5C; Toccalino et al., 2016 Figure 3). We did not observe a significant attenuation of the effects of social context on sequencing in these birds with atropine, but the conclusions that can be drawn from this are clearly limited given the weak modulation at baseline. We have presented the relevant data in Figure 6—figure supplement 1 and discussed these caveats in the Results section and in the Materials and methods section (see also our responses to Reviewer #1, point #5 and Reviewer #3, point #1).

We agree with Reviewer #2 that it would be interesting to examine how HVC activity around ‘branch points’ in song is altered by carbachol and directed song. However, we have not attempted to add such a description, as we think it is beyond the scope of what we could reasonably incorporate into the current manuscript. This is largely because there is currently little understanding of how activity in HVC relates to variable transitions even in the absence of any manipulations. Hence, extensive additional data, analysis, and description would be required to appropriately contextualize any findings regarding the effects of carbachol or social context (see also our response to Reviewer #2 for more commentary on this point).

3) Please provide a finer grain analysis of the durations of individual syllables and gaps in directed vs. undirected song as opposed to the overall length of predominant motifs.

The revised manuscript now includes an analysis of syllable and gap durations as requested. We found that both syllables and gaps were shortened by carbachol and directed song, but that the effects were larger for gaps than for syllables (Figure 1—figure supplement 1 and Figure 6—figure supplement 1; see also our response to Rreviewer #1, point #4 for more detail).

Reviewer #1:[…]1) Acetylcholine is of course involved in arousal, and the authors have good evidence that this pathway is activated during the arousal that accompanies directed song. Nevertheless, the cholinergic system is also a major participant in circadian patterns of arousal and alertness. In light of the sizable body of data that show circadian variability in song behavior and neural activity in the song system, some speculation in the discussion about how their data might interface with that line of work seems warranted.

This is a great point, and we agree that some discussion of circadian changes in songbird behavior and neural activity is warranted. We now comment on these circadian changes in the Discussion section, in particular with respect to the ascending arousal system. More specifically we note that some of the previously described circadian variation in neural activity and song could reflect changing activity in the ascending arousal system, including cholinergic components that have been linked to changes in wakefulness and alertness (see subsection “Contributions of HVC and the cholinergic system to social modulation of song”).

2) I recommend the authors add citations of these relevant papers to their Introduction: Neuron 46:173; Cerebral Cortex 20:2739.

Thanks – we agree that these papers pertaining to the role of the cholinergic system in motor skill learning are relevant, and we now reference them in the second paragraph of the Introduction.

3) I understand why the authors limited their pitch analysis to syllables with defined harmonic structure that were also constant in pitch, but the authors might consider analyzing a subset of different syllables that don't share these attributes to see if an increase in pitch is global to song or if noisier or modulated syllables are affected differently. For example, syllables that contain substantial spectrotemporal modulation may show an increase in the span of that spectrotemporal modulation rather than a uniform upward shift in pitch.

We found that it was difficult to assess changes in pitch (fundamental frequency) for syllables with frequency-modulated harmonic components (“sweeps”), due to the increase in tempo produced by carbachol. Pitch for a given syllable was analyzed by averaging the pitch contour over a short time window in which the fundamental frequency is relatively constant; for harmonic stacks, with relatively constant fundamental frequency, pitch measurements are robust to variation in the timing of the measurement window. In contrast, for frequency-modulated sweeps, small variation in the timing of the measurement window can result in inaccuracies in estimation of pitch. For example, for a downward sweeping syllable, a slightly later measurement window within the syllable results in a lower estimate of pitch. This is especially problematic for sweep syllables in which there has been an increase in tempo, such that differences in pitch between two renditions will vary depending on whether a measurement window is aligned relative to the onset or offset of the syllable. Indeed, we found that subtle differences in measurement procedures for sweep syllables could result in variation in pitch measurements that was large compared to the small systematic shifts in pitch that were observed for harmonic stacks (~1.5%). Nonetheless, we agree that it is worthwhile to investigate whether the behavioral effects of carbachol are observed for other syllable types in the bird’s repertoire. To investigate this, we focused on amplitude changes caused by carbachol, as this is a syllable feature that was strongly modulated by carbachol (~8%), and because amplitude can readily be measured for all syllable types. We found that carbachol induced comparable increases in amplitude for syllables of all types (harmonic stacks, sweeps, complex syllables, and noisy syllables). We present this analysis in Figure 1—figure supplement 1C and associated text in subsections “Pitch and pitch variability” and “Amplitude”).

4) I'm curious why the authors did not separately analyze the duration of each syllable and silence between them, instead choosing to analyze the overall duration of common groups of syllables. As far as I know, this is not the approach taken by past studies of contextual modulation of song behavior.

In our initial submission, we analyzed the duration of syllable sequences (from syllable onset to syllable onset) because the shape of the amplitude envelope at syllable onsets is less variable than for offsets, yielding more accurate estimates of tempo. Previous studies of social modulation of song have also assessed changes in song tempo by measuring sequence/motif durations, though some have measured sequences from syllable onset to syllable offset (Aronov and Fee, 2012; Cooper and Goller, 2006; Sakata et al., 2008). In our revised manuscript, we have added an analysis of syllable and gap durations for both the carbachol experiments (Figure 1—figure supplement 1) and the social context experiments (Figure 6—figure supplement 1). In both cases, syllables and gaps were each shortened, with a larger effect on gap durations, which as we now note in the main text parallels a similar differential effect on gaps versus syllables caused by cooling of HVC.

5) For me, one of the most interesting results was the experiment depicted Figure “. The authors' finding that a manipulation of HVC alone affected song sequencing is quite noteworthy. As far as I understand, the neural mechanisms of song sequencing, particularly in species with significant sequence variability like Bengalese finches, remain controversial. This result seems to support the idea that song sequencing is entirely contained within the HVC circuit. Therefore, I'm not sure these data deserve to be relegated to a supplementary figure. That said, if the authors were to move it to a main figure, they should probably do a more comprehensive analysis of sequence changes. I think here the focus on the most common sequences, which like in point #1 above, may be affected differently than rarer sequences.

We agree that the sequencing results are interesting and merit further elaboration. As suggested, we now report sequencing results in a main figure (Figure 2), and provide additional characterization of these effects. Our results show that carbachol significantly alters the probabilities of different transitions at branch points (Figure 2A-C) and significantly alters syllable repetitions (Figure 2D-F). We also now report in the Results section that these sequencing changes are associated with a trend towards reduction in “transition entropy” of the sort that has been observed for directed song. As we note in our response to Reviewer #3 (point #1), we also assessed potential contributions of acetylcholine to social modulation of song sequencing (Figure 6—figure supplement 1).

Regarding the broader question of the role of HVC in syllable sequencing in the Bengalese finch, while our data indicate that carbachol dialyzed into HVC can significantly alter syllable sequencing (consistent with prior data from Zhang et al., 2017 on the effects of cooling Bengalese finch HVC), they do not rule out the possibility (likely in our view) that syllable sequencing is additionally influenced by activity elsewhere in song system, including via recurrent projections from the brainstem back to HVC.

6) I noticed something in Figure 2B that may be interesting if it is consistently observed. In these examples, the carbachol experiment in which LMAN is intact shows a larger pitch shift in earlier songs that relaxes partially in later songs. This is not seen in the example they show in which LMAN has been inactivated. Was this something they observed consistently? Because it might suggest that after many songs performed in the presence of carbachol, the bird begins to correct or compensate for the pitch shift, and that this may be LMAN-dependent.

Thanks for pointing this out – this is a thoughtful observation that we agree merits further investigation, so we took a look. We did occasionally observe transient behavioral effects following carbachol infusion – lasting on the order of tens of minutes – prior to a relatively stable behavioral response that persisted for the remainder of the experiment (such as the example you refer to in Figure 3B, previously Figure 2B). However, these transient effects were not observed consistently, even within birds in which we conducted multiple experiments (see the example experiment in Figure 1C). Addressing your question more directly, we occasionally observed this pattern of strong early increase followed by partial relaxation in the carbachol + LMAN inactivation experiments. Thus, it seems that this transient response is not strictly LMAN dependent. We chose to analyze behavioral effects in a time window beginning one hour after drug onset, primarily to ensure that carbachol would have time to diffuse throughout HVC. However, this time window also corresponds to a point at which other idiosyncratic transient effects have settled. We now make note of these points in “Definition of analysis windows” under the subsection “Analysis of song features” in Materials and methods.

7) I'm not sure that the authors assertion that HVC firing rates are specifically elevated just before the syllable onset is very well justified by the data. Although there is a difference in Figure 3E between firing rates just before and just after the start of the syllable, there is still a large increase in both in the presence of carbachol. Also, why did they not perform this temporally limited analysis on the data in Figure 4?

Thanks for catching our confusing wording. We did not mean to imply that the increase in firing rates was restricted to the period just prior to syllable onsets, and we have revised our wording in the main text to make this clear. We now specifically note that carbachol caused an increase in firing rates in windows both preceding and following syllable onset, with a modestly larger effect in the premotor window. We now also report the results of this temporally-limited analysis for the directed and undirected song HVC recordings (see Figure 5D and associated text of theResults section). Relative to undirected song, directed song was associated with a significant increase in firing rate both preceding and following syllable onsets. The magnitude of this increase did not differ significantly between the pre and post-syllable onset windows.

8) There are at least three major classes of neurons in HVC based on their projection targets: RA-projecting, X-projecting, and interneurons. The projection neurons fire much less frequently than the interneurons, so their multiunit recording data is almost certainly overwhelmingly dominated by the activity of interneurons. This is an important point that in my opinion, they have to address, both because the activity the authors see likely doesn't reflect much of the activity of the pathway they think is mediating the effects of carbachol, and also because the interneurons are GABAergic and probably target RA-projecting neurons with synaptic inhibition. This leads to potentially a different interpretation of the changes in HVC firing rate that they see in carbachol infusion and directed song.

We agree that multi-unit activity in HVC is likely to be dominated by inhibitory interneurons, and have added further discussion regarding possible ways in which cholinergic modulation of HVC activity could contribute to the observed behavioral changes (see the subsection “Neural mechanisms underlying the control of motor vigor in songbirds”). In this section we note that the reasonable intuition that an increase in the firing rate of interneurons should result in a decrease in the firing rate of projection neurons is likely too simple. For example, we note that both projection neurons and interneurons increase their activity during song relative to non-singing quiescent periods (Kozhevnikov and Fee, 2007). Additionally, we now discuss the possible circuit level effects of acetylcholine on HVC during singing more explicitly in relation to the previously reported cellular effects of acetylcholine in acute HVC slice recordings (Shea et al., 2010).

In particular, we note that muscarinic agonists applied in vitro tend to hyperpolarize interneurons. If interneuron firing rates are nevertheless increased by cholinergic agonists applied in vivo, one possibility is that any hyperpolarization at a cellular level is overcome by increased excitatory input to interneurons. We hypothesize that such increased excitatory input could derive from HVC projection neurons, which are depolarized by muscarinic agonists in vitro (Shea et al., 2010), and which synapse on HVC interneurons. In the Discussion section noted above, we now provide a more extended commentary on the implications of our findings in light of these prior studies.

Reviewer #2:[…]A comment on analysis:"Thus, the increased pitch stereotypy caused by carbachol cannot be explained by reduced neural variability in HVC."The authors do not really return to this point in the Discussion section, but I was interested to understand why/how this dissociation might be relevant to theories about syllable production and variability in pitch that is exploited in many reinforcement paradigms.

In response to a comment from Reviewer #3, we have tempered our wording and conclusions on this point to reflect that while variability was not reduced in our multi-unit recordings, this does not rule out the possibility of a reduction in the variability of projection neurons. Nonetheless, we agree that the reduction in behavioral variability following a manipulation of HVC is noteworthy in its own right. Most prior work emphasizes the contribution of the AFP to acoustic variability (e.g. Hampton et al., 2009; Kao et al., 2005; Leblois et al., 2010; Stepanek and Doupe, 2010). Our findings suggest that a significant source of behavioral variability originates within HVC. Conceivably, this variability could be harnessed in the service of reinforcement learning in much the same way that is thought to occur for variability originating from the AFP (Charlesworth et al., 2012). Our revised manuscript now includes a brief discussion of this point, as well as a discussion of potential neural mechanisms underlying the reduction in behavioral variability (see the subsection “Contributions of HVC to acoustic variability”).

The transition probability effect at branch syllables was not, as far as I could find, addressed in light of recordings from HVC. It seemed however that since HVC activity sequences are thought to determine transition probabilities this would be quite interesting to explore. Was it not possible to observe HVC activity specifically around branch syllables for sampling reasons? Or were there other reasons this was not explored in more detail? Does social modulation also modulate transition probabilities?

We agree that HVC neural activity around branch points could be quite revealing with respect to the neural mechanisms underlying the effects of carbachol on song sequencing. In response to this suggestion, we considered the possibility of adding analysis of neural activity at branch points and how this activity is influenced by experimental conditions (i.e. carbachol or directed song). However, we concluded that the addition of neural data that would be required to describe and contextualize any findings is beyond the scope of what we could reasonably incorporate in an expanded manuscript. This is in part because addressing how neural activity at branch points is affected by various manipulations would require an initial description of normal or “baseline” activity at branch points, which itself is a complex enterprise that we think will ultimately merit its own story. For example, a baseline description of sequencing under saline conditions would need to address questions such as: When and how do activity patterns for different possible transitions diverge with respect to syllable boundaries? How heterogeneous are these dynamics across different recording sites and different branch points? And are these dynamics related in any systematic way to transition probabilities? In short, a characterization of how carbachol and directed song affect HVC activity around branch points would require extensive characterization of this activity in the absence of any manipulations.

Regarding whether social context modulates transition probabilities, previous studies have shown that social context can modulate transition probabilities for both “branch points” and repeated syllables in Bengalese finch song (see for example Sakata et al., 2008), though such effects have not always been observed, indicating that they are less robust than effects on syllable structure (e.g. Hampton et al., 2009; Toccalino et al., 2016). Our revised manuscript now provides a more thorough characterization of experimental influences on syllable sequencing for both carbachol and social context experiments (Figure 2 and Figure 6—figure supplement 1). See also our responses to Reviewer #1, point #5 and Reviewer #3, point #1 for more detail.

A general point:At least in mammals ACh also acts within BG via intrinsic cholinergic neurons in striatum and globus pallidus as well as projections from midbrain (PPN for example) on to midbrain dopamine neurons. In the design of the current experiments the coordination of ACh signaling mediated by these pathways might be lost. It would be useful to discuss the point that while the Ach-mediated vigor mechanisms can be dissociated from putative dopamine-dependent mechanisms using local infusion of carbachol, the two mechanisms may in fact be coordinated in other conditions. Perhaps consistent with this in Figure 5 it would appear that HVC-infused atropine mediates only part of the social modulation of song consistent with possibility that mAChR function in other (possibly BG-related) brain areas could contribute to further effects. This might make additional infusions of atropine in the context of directed song interesting to explore whether cholinergic modulation more broadly than in HVC is necessary to account for the full extent of social modulation.

Thanks – we have revised the Discussion section to emphasize that these two pathways may indeed be coordinated in many situations, making note of some of the subcortical anatomy that could serve as an anatomical substrate (see the subsection “Distributed circuits for the control of motor vigor”). We agree that it would be interesting to explore whether cholinergic signaling in other brain regions contributes to social modulation of song and expect that future studies will investigate this topic.

Comments on text/interpretation:"the extent to which acetylcholine contributes to enhanced motor vigor observed in aroused behavioral states remains unknown."Maybe modify this point a bit. there is a certainly a strong association between cholinergic activity and aroused behavioral states observed in mammals (Jones, 2004) and there has long been data on how mAChR antagonists effect aroused behavioral responses (such as conditioned responses, e.g. Longo, 1966).

Sorry for the confusion – we intended to refer in particular to cholinergic signaling in motor cortical regions. We have revised the sentence referenced here to make this more explicit: "However, while motor cortical regions receive dense cholinergic innervation from the nucleus basalis (NBM; Eckenstein et al., 1988; McKinney et al., 1983; Raghanti et al., 2008), the extent to which cholinergic signaling in cortex contributes to motor invigoration observed in aroused behavioral states remains unknown."

"Moreover, for each feature, the changes elicited by combined carbachol + LMAN inactivation were not significantly different from the sum of the individual effects of carbachol and LMAN inactivation (pitch: p = 0.77, pitch c.v.: p = 0.85, tempo: p = 0.22, amplitude: p = 0.71, signed-rank test). These results indicate that increased cholinergic tone in HVC can modulate song via primary motor circuitry independently of input from the songbird basal ganglia."I thought this result was a particularly key demonstration that these two pathways appear to linearly sum. I would note that a linear sum of BG-independent and BG-dependent mechanisms has been proposed previously (Yttri and Dudman, 2018) – although this provides the clearest direct evidence for ~linear combination of effects to date.

Thanks for pointing this out – we now make reference to the model described in the Yttri and Dudman, 2018 paper in the Results section for these experiments, and also in the Discussion section as an example of prior work that has recognized contributions of cortical circuits to the control of vigor (see the subsection “Distributed circuits for the control of motor vigor”).

"Prior work has identified basal ganglia circuitry as an important locus for the control of motor vigor (Panigrahi et al., 2015; Schmidt et al., 2008; Yttri and Dudman, 2016). In principle, cortical and basal ganglia circuitry could jointly control movement vigor, with coordination between these pathways mediated by bidirectional feedback between them (Bosch-Bouju et al., 2013). However, our findings indicate that in some situations, modulation of motor vigor can occur independently of basal ganglia circuitry."The Discussion section as written suggests that previous treatments have suggested that basal ganglia (BG) might be an exclusive determinant of motor vigor. However, I would just note that this was explicitly presented as an argument for BG-independent pathways for control of movement vigor previously – as noted in Dudman and Krakauer (2016) cited in this manuscript: "It should be stressed that just because the basal ganglia can influence vigor this does not imply the converse: that vigor parameters are always under the obligate control of the basal ganglia." This is also explicit in normal treatments of this model of vigor in the equations in Yttri and Dudman (2018) in which BG-independent component of vigor sums (is independent of) the BG contribution.Nonetheless, I think the authors make a very important point here that an explicit modulatory influence that does function in cortex independent of BG to control vigor is a very valuable addition/demonstration. The authors might be interested to consider other phenomena that are similarly hard to reconcile with an exclusive reinforcement mechanism like verbally-instructed changes in movement vigor in normal subjects ("Move faster!"), or the learned changes in vigor present that persist in dopamine depleted animals (Panigrahi et al. 2015) and patients (Mazzoni et al., 2007; Baraduc et al., 2013), or as authors do discuss paradoxical kinesia. (These were the observations that led to the proposal ofBG-independent circuits for motor vigor in previous reviews). Moreover, similar to the bird circuitry highlighted in the discussion of this manuscript, in the mammal BG-output also converges on subcortical targets of descending motor cortical projections making these parallels perhaps even closer.

Thanks for bringing this work to our attention – we agree that much of this is relevant, and have incorporated in particular the possibility that cholinergic signaling in cortex could contribute to learned changes in motor vigor. As you point out, verbally-instructed changes in vigor and paradoxical kinesia may also rely on non-basal ganglia circuitry, and these are points that we note in the Discussion section (see the subsection “A potential role for the cholinergic system in movement disorders”).

Reviewer #3:[…]1) Historically, HVC has been thought to be important for tempo and sequencing while the AFP contributes to syllable structure. This makes Bengalese finches an interesting model to study HVC because they have greater sequence variability than zebra finches (a focus that Brainard has taken advantage of in the past). In particular, his lab has found that during directed singing, sequence entropy decreases and syllable repeats increase and this has not been found to be affected by lesions of LMAN, thus hinting that HVC may be significant in this modulation. Here, they report that Ach manipulation in HVC affects sequence transitions (though there is little detail on this change, including whether there is a change in entropy) and also increases repeats (similar to the effect of directed singing). However, there is no further discussion of sequence or repeats through the rest of the paper. I would like to see the effects on sequence and repeats throughout. It would be especially interesting to know whether atropine affects the decrease in entropy and increase in repeats previously reported for directed singing. Presumably, this would not require additional experiments as these data would be available in the data already collected.

As requested, we have now analyzed how carbachol affects transition entropy; we observed a trend toward reduced sequence entropy similar to directed song, though this was not statistically significant (see the text of the Results section for these experiments). Following a suggestion from Reviewer #1 (point #5), the sequencing effects from the carbachol experiments are now reported in a main figure (Figure 2).

We found that it was difficult to definitively determine whether or not changes in sequencing observed during directed song are dependent on cholinergic signaling in HVC, due to the small number of repeated syllables and branch points in the dataset, the limited number of songs we had available to estimate transition probabilities, and possibly instabilities in sequencing effects observed across repeated sessions of directed singing (as reported in Hampton et al., 2009). In particular, the effects of social context on sequencing were not very robust in the subset of birds tested with manipulations of social context and concurrent dialysis of atropine into HVC. Only two out of five repeated syllables in these birds exhibited a significant increase in syllable repetitions during directed song, and this increase was not attenuated by atropine (Figure 6—figure supplement 1C). We did not observe a significant decrease in transition entropy during directed song for these birds, so that there was not a meaningful opportunity to look for attenuation by atropine. Similarly, we did not observe an attenuation of the change in branch point transition probabilities with atropine (Figure 6—figure supplement 1D).

2) I'd like more raw data and interpretation of the multi-unit recordings. It's unclear what these sites look like (how multi-unit is multi-unit?). Moreover, I find the rasters to be a poor way to convey the effects. In the one in Figure 3, there is so much black that you notice the decrease in white-space more than the increase in black, which gives the impression of a decrease rather than an increase. In Figure 4 I can't see the difference in the rasters. For interpretation, is the idea that there are just more neurons firing during directed singing, or that there is an increase in the rate of individual neurons, or both? Work using IEG expression in Bengalese finches indicates that there are more EGR1 expressing cells during undirected singing that directed singing, which would appear to be at odds with the current multi-unit result, but this is not discussed.

As requested, the revised manuscript now includes substantial additional characterization of our multi-unit neural recordings and more interpretation in the Discussion section (see also our response to Reviewer #1, point #8). We now show a number of examples of the raw data from these recordings (Figure 4 and Figure 4—figure supplement 1). Additionally, we now report the distribution of firing rates (Figure 4—figure supplement 1A) and signal-to-noise ratios (Figure 4—figure supplement 1B and 1C) for all recording sites, which speak to the question of how 'multi-unit' the recordings are.

We agree that it was difficult to see firing rate changes induced by carbachol or social context in raster plots due in part to the density of points. We have enlarged the raster plots in Figure 4 (previously Figure 3) to decrease the density of points so that the modulation of activity might be more apparent. Nevertheless, we would agree that the effects of carbachol and of social context are more apparent in the averaged firing rates presented in these figures. We still think that the raster plots are useful in part to give a visual impression of the raw data with respect to song locked modulation of activity and overall stability of the pattern of neural firing, and have therefore retained them in the figures. For example, the raster plots illustrate that HVC activity is strongly modulated during song and that the pattern of this modulation is remarkably consistent from trial-to-trial, a typical observation for HVC activity that otherwise may not be apparent to those outside of the birdsong field.

We agree with the implicit comment regarding limitations on the interpretation of changes to multi-unit firing rates; it is difficult to make strong inferences from the observed increases in multi-unit activity about underlying changes to the activity of individual neurons – this includes the question of whether any formerly silent neurons are recruited, as well as the question raised by Reviewer #1 of how the different neural subtypes within HVC are affected. In our revised Discussion section we address the issue of how changes to HVC activity across different neural populations could contribute to the observed behavioral changes, but are careful to note that our models are speculative (see also our responses to Reviewer #1, point #8).

We have expanded the discussion of how our results might relate to the previous observation that IEG expression is modulated by social context (Matheson et al., 2016). We first note that the two studies are consistent in indicating that HVC activity is modulated by social context. We then discuss various differences between neural recordings and IEG expression that could reconcile our observation of an increase in multi-unit activity during directed song with an observed decrease in IEG expression during directed song. These include potential differences arising from the long integration time of the IEG response, the nonlinear relationship between neural activity and IEG expression, the specific neural types examined, and other aspects of experimental design (see the subsection “Contributions of HVC and the cholinergic system to social modulation of song”).

3) There are a number of instances in which there are many syllables and/or recordings sites in a single bird but there is no indication that bird has been included as a random variable in the statistical models. It is actually quite important to include bird ID as a variable in the model to account for the fact that many measurements have been made in the same individual (or to indicate more explicitly if it has already been included).

We agree. In particular, we note that while we (and many other birdsong publications) implicitly assume that all syllables can be treated as independent measurements, this is not formally correct, since different syllables sampled from the same bird may exhibit correlated effects, resulting in a form of pseudoreplication that could inflate estimates of statistical significance. As suggested, we can account for this type of structured data by including the identity of each bird as a random effect in a linear mixed effects model. Our revised manuscript now includes statistical analysis based on mixed effects models to supplement the original statistical tests. The significance tests of our main results are essentially unchanged. Details are provided in the Materials and methods section and in supplementary files that accompany the main figures (see the subsection “Mixed effects models” under “Statistics”).